# Somatic CpG hypermutation is associated with mismatch repair deficiency in cancer

Aidan Flynn[1,2,3], Sebastian M Waszak [ID][4,5,6✉] & Joachim Weischenfeldt [ID][1,2,7,8✉]

## Abstract

**Somatic hypermutation in cancer has gained momentum with the increased use of tumour mutation burden as a biomarker for immune checkpoint inhibitors. Spontaneous deamination of 5-methylcytosine to thymine at CpG dinucleotides is one of the most ubiquitous endogenous mutational processes in normal and cancer cells. Here, we performed a systematic investigation of somatic CpG hypermutation at a pan-cancer level. We studied 30,191 cancer patients and 103 cancer types and developed an algorithm to identify somatic CpG hypermutation. Across cancer types, we observed the highest prevalence in paediatric leukaemia (3.5%), paediatric high-grade glioma (1.7%), and colorectal cancer (1%). We discovered germline variants and somatic mutations in the mismatch repair complex MutSα (*MSH2-MSH6*) as genetic drivers of somatic CpG hypermutation in cancer, which frequently converged on CpG sites and *TP53* driver mutations. We further observe an association between somatic CpG hypermutation and response to immune checkpoint inhibitors. Overall, our study identified novel cancer types that display somatic CpG hypermutation, strong association with MutSα-deficiency, and potential utility in cancer immunotherapy.**

**Keywords** Pan-Cancer; CpG Hypermutator; MMR; TMB; Immunotherapy
**Subject Categories** Cancer; Chromatin, Transcription & Genomics

## Introduction

Mutations in cancer genomes can be inherited or acquired somatically during tumour evolution. The tumour mutation burden (TMB) is the sum of somatic mutations that arose due to intrinsic cell-type-specific mutational processes and exposure to endogenous and exogenous mutagens (Pleasance et al, 2010). While most cancers have modest TMBs, certain tumours can acquire a hypermutator phenotype associated with higher TMB. Hypermutated tumours are often distinguished through cohort-specific outlier analyses, which have found hypermutator thresholds of 10–20 mutations per Mb (Campbell et al, 2017; Chalmers et al, 2017). In a recent pan-cancer whole-genome study of more than 2700 primary, treatment-naïve tumour samples, hypermutated tumours represented 5% of all tumour genomes and contained more than half of all the mutations (Alexandrov et al, 2020). Hypermutated tumours are predictive of increased sensitivity towards immune checkpoint inhibitors (ICI) (Samstein et al, 2019; Le et al, 2017; Rizvi et al, 2015), likely due to expressed neoantigens that can be recognised through MHC class I presentation on the tumour cells. Whether and to what extent specific properties of hypermutated tumours predict response to ICI is currently unclear.

Hypermutation can be caused by extrinsic mutagens such as UV exposure in melanoma and tobacco exposure in lung cancer (Alexandrov et al, 2013; Campbell et al, 2017). Intrinsic mutational processes such as dysregulation of DNA damage response and processing enzymes can act as cancer drivers to cause somatic hypermutation. Error-prone DNA replication is a significant source of mutations and is estimated to be associated with two-thirds of all mutations in cancer genomes (Tomasetti et al, 2017). The canonical DNA mismatch repair (MMR) pathway operates downstream of the replication fork to correct mis-incorporated bases through the activity of the MutSα complex proteins MSH2 and MSH6. The MutLα complex proteins MLH1 and PMS2 are recruited jointly with EXO1 to remove base-base mismatches and the surrounding nucleotides (Kunkel and Erie, 2015). MMR deficiencies (MMRd) are observed in both inherited and sporadic cancers and are associated with a higher TMB.

MMRd causes not only somatic hypermutation but also leaves characteristic patterns of mutations (often referred to as 'mutational signatures') in cancer genomes (Alexandrov et al, 2013). MMRd has been associated with several single-base substitution (SBS) signatures (Alexandrov et al, 2020), including SBS 6, 15, 21, 26 and 44. Moreover, several hypermutation-associated signatures have been identified, and they are primarily thought to occur during DNA replication. In contrast, the most dominant mutational signature across all cancer genomes is SBS1, associated with $C > T$ mutations at CpG dinucleotides, and is thought to be caused by spontaneous deamination of 5-methylcytosine (5mC). SBS1 is also termed 'clock-like' because it has been observed to increase with age at diagnosis (Alexandrov et al, 2015). All cells

[1]Biotech Research & Innovation Centre (BRIC), University of Copenhagen, Copenhagen, Denmark. [2]The Finsen Laboratory, Copenhagen University Hospital - Rigshospitalet, Copenhagen, Denmark. [3]Department of Clinical Pathology and Centre for Cancer Research, University of Melbourne, Parkville, VIC, Australia. [4]Swiss Institute for Experimental Cancer Research (ISREC), School of Life Sciences, École Polytechnique Fédérale de Lausanne (EPFL), Lausanne, Switzerland. [5]Centre for Molecular Medicine Norway, Nordic EMBL Partnership, University of Oslo and Oslo University Hospital, Oslo, Norway. [6]Department of Neurology, University of California, San Francisco, San Francisco, CA, USA. [7]The DCCC Brain Tumor Center, Danish Comprehensive Cancer Center, Copenhagen, Denmark. [8]Department of Urology, Charité University Hospital, Berlin, Germany. ✉E-mail: sebastian.waszak@epfl.ch; joachim.weischenfeldt@bric.ku.dk

experience spontaneous hydrolytic deamination of purines and pyrimidines in a replication-independent manner, with a particularly high rate of 5mC in a CpG sequence context (Bird, 2002). The majority of CpG sites in the genome are methylated, and deamination of 5mC to thymine causes G:T mismatches, which lead to C > T mutation during DNA replication if not recognised and repaired by MMR. Repair of post-replicated DNA lesions occurs through MMR of the newly synthesised strand by leading and lagging-strand polymerases (Pol-ε and Pol-δ, respectively, reviewed in (Cortez, 2019; Kunkel, 2009).

The alternate, non-canonical MMR (ncMMR) acts in a replication-independent manner and was first identified in somatic hypermutation of immunoglobulin genes (Martomo and Gearhart, 2006; Peña-Diaz et al, 2012) and also involved in repairing DNA lesions in transcribed genes, facilitated by active transcription and the histone modification H3K36me3 (Huang et al, 2018). Hydrolytic deamination of methylated Cytosine is usually repaired by the error-free base-excision repair (BER) pathway involving the glycosylase MBD4, but the repair intermediates can be 'hijacked' by ncMMR, resulting in an error-prone repair process (Chen and Furano, 2015; Fang et al, 2021). We and others have recently demonstrated a genetic association between pathogenic germline variants in *MBD4* and somatic C > T mutations at CpG dinucleotides in cancer (Sanders et al, 2018; The ICGC/TCGA Pan-Cancer Analysis of Whole Genomes Consortium, 2020).

Here, we undertook a pan-cancer mutation study of 30,191 cancer patients across 103 different tumour types to expand and explore genetic associations with CpG-hypermutated tumour genomes and the impact on driver genes during tumour evolution. We examined the implications of somatic CpG hypermutation for understanding the 'clock-like' mutational signature, the emergence of driver mutations, and implications for patients treated with immune checkpoint inhibitors.

# Results

## Discovery of a somatic CpG hypermutator phenotype in cancer

Pan-cancer analysis of TMB has identified several histological types with an increased incidence of hypermutated tumours (Campbell et al, 2017; Chalmers et al, 2017). Yet, somatic hypermutation in a CpG sequence context (i.e. CpG>TpG) has not been assessed across paediatric and adult cancer types. To this end, we obtained somatic mutations from 195 previously published exome and whole-genome sequencing studies covering 105 histological subtypes (Methods, Data ref: https://github.com/cBioPortal/datahub, v2.11.0, Data ref: https://pedcbioportal.kidsfirstdrc.org, accessed November 2019), the ICGC/TCGA Pan-Cancer Analysis of Whole Genomes (PCAWG) Synapse repository (Data ref: https://www.synapse.org/#!Synapse:syn11801870, accessed November 2019; Data ref: https://dcc.icgc.org/pcawg, accessed April 2019; Data ref: http://www.synapse.org/glass, data release version 2019-03-28)). To avoid bias from small sample sets, we removed cohorts with less than 20 samples where a cohort was defined as a group of samples from the same study with a shared histological type and sequencing method. The final dataset represented 58.8 million somatic mutations across 30,950 tumour samples and 30,191 patients (Fig. 1A). To systematically identify tumours with CpG

hypermutation, we required that the tumour should be an upper outlier in both the number of somatic mutations and the proportion of mutations in a CpG context (Fig. 1A–C). We used Tukey's rule to identify hypermutated (HM) tumours within their respective histological type and study (group outlier threshold, Fig. 1B,C). In addition, we required outlier tumour samples to also have a tumour mutation burden greater than the median across all samples in the entire cohort (1.33 mutations/Mb), to prevent bias toward cohorts consisting of tumours with a low average mutation burden. Similarly, we used Tukey's rule to compute an upper outlier threshold for the proportion of C > T mutations in a CpG context across all samples, which resulted in a global CpG hypermutation outlier threshold of 0.60 (i.e. C > T mutations at CpG per total number of mutations, Fig. 1B,C).

We identified 1914 (6.3%) patients with somatic hypermutation (HM) across 93 cancer types based on these criteria. Of these, 76 patients across 23 cancer types fulfilled our requirements for a somatic CpG hypermutation phenotype (HM$_{CpG-Hi}$) (Fig. 1D; Table EV1). We estimate that 1 in 397 cancer patients developed a tumour with a CpG hypermutator phenotype. We next sought to identify whether the CpG hypermutation phenotype was compatible with any known SBS signatures. To this end, we separated the cohort into HM tumours above and below our CpG outlier threshold, HM$_{CpG-Hi}$ and HM$_{CpG-Lo}$, respectively, and performed a separate SBS mutational signature contribution analysis. Each signature was compared to the official COSMIC mutational signatures (Methods) using cosine similarity (a cosine score larger than 0.9 is considered a perfect similarity (Degasperi et al, 2020; Omichessan et al, 2019). Both groups were dominated by C > T mutations, as expected. The HM$_{CpG-Lo}$ group was composed of different signatures, none of which reached the cosine similarity cut-off of 0.9 for a signature match (Fig. 1E iii-iv). The CpG hypermutated tumours, on the other hand, displayed a strong similarity to SBS1 (cosine 0.95). SBS1 is associated with spontaneous deamination of 5-methylcytosine (5mC), resulting in C > T substitutions, and we have recently associated SBS1 with base-excision repair and *MBD4*-deficiency in pan-cancer studies (The ICGC/TCGA Pan-Cancer Analysis of Whole Genomes Consortium, 2020). We further confirmed somatic CpG hypermutation in all 78 tumours (76 patients) using an independent enrichment-based approach (Roberts et al, 2013; The ICGC/TCGA Pan-Cancer Analysis of Whole Genomes Consortium, 2020) (Fig. EV1; Table EV2). Having classified CpG hypermutated tumours with a signature indistinguishable from the clock-like SBS1, we next explored cancer type-specific patterns.

## Somatic CpG hypermutation frequently occurs in paediatric malignancies and colorectal cancer

We applied an exact binomial test to each histological type to assess whether CpG hypermutation was observed more frequently than expected. Of the histological types that reached significance, the highest number of CpG hypermutation samples was detected in colorectal cancer (1.1%, 18/1660) (Fig. 2A; Table EV1). This cancer type is strongly linked with MMRd due to somatic or germline mutations in *MLH1*, *MSH2*, *MSH6*, *PMS2*, and *MLH1* promoter hypermethylation, and enhanced levels of MMRd-associated mutational signatures (e.g., SBS6). Brain tumours with a CpG hypermutation phenotype (0.53%, 15/2727) were predominantly

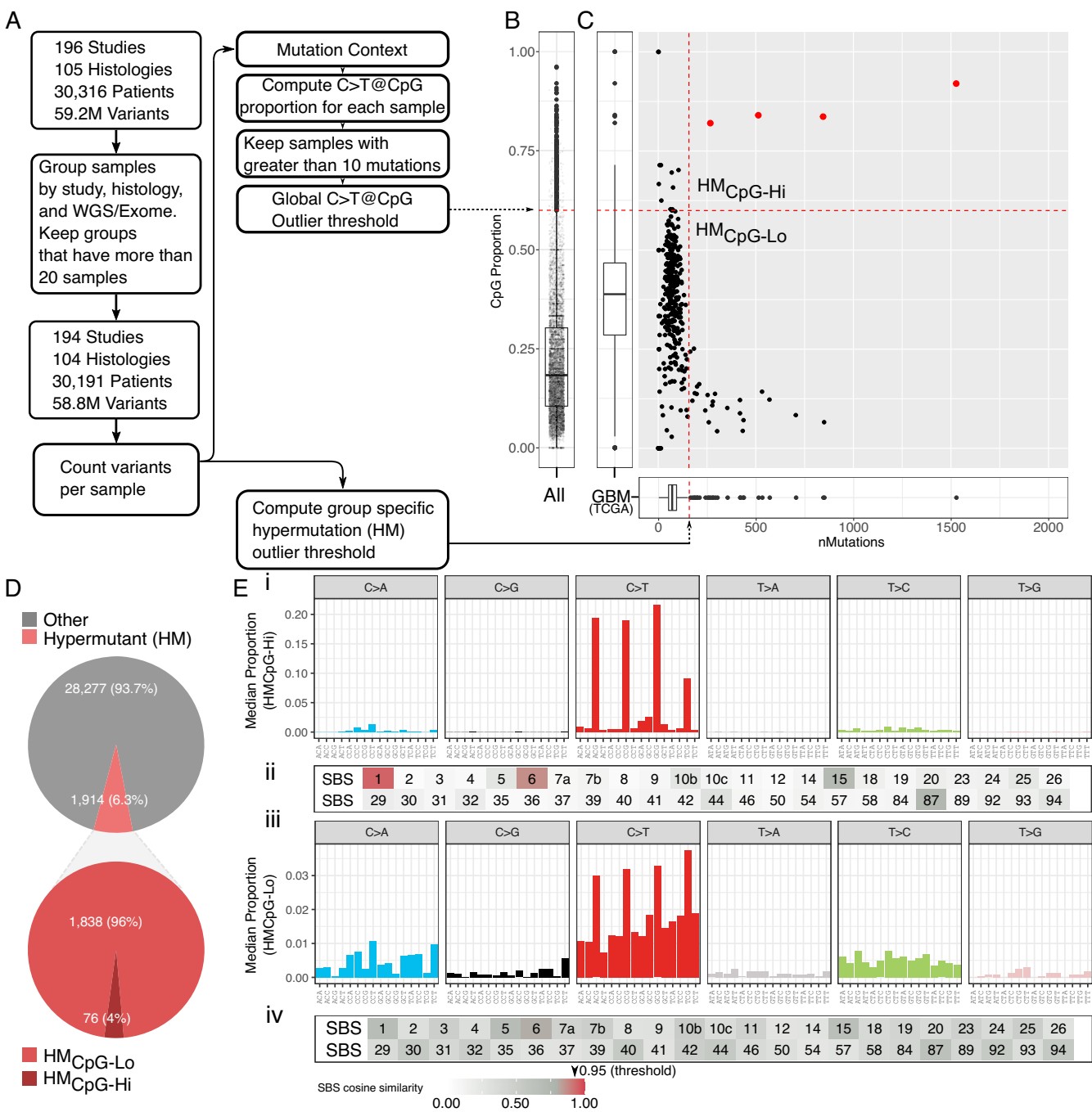

**Figure 1. Discovery of somatic CpG hypermutation in cancer.**

(A) A schematic of the workflow used for computing upper outlier thresholds for identifying samples with a high mutation burden and a high proportion of C > T mutations in a CpG context (Np[C > T]pG). (B) A box and scatter plot showing the distribution of the proportion of mutations that were Np[C > T]pG for all samples in the cohort with greater than ten mutations ($n = 25{,}805$). An outlier threshold was computed from the 75th percentile (0.305, median: 0.185) and interquartile range (0.193, data range: 0–0.96) using Tukey's fence (dashed red line). (C) An example calculation of the tumour mutation burden threshold for a single study (glioblastoma, TCGA-GBM, $n = 392$) using Tukey's fence (vertical dashed red line) and the application of the cohort-wide cutoff for the proportion of Np[C > T]pG mutations (horizontal dashed red line). Red dots represent samples determined to be outliers by both thresholds. (D) Relative breakdown of the percentage of patients within the cohort separated by our hypermutant (HM) cut-off (HM and non-HM, pink and grey section, respectively). HM patients are further divided by our global Np[C > T]pG outlier threshold (CpG-Hi and CpG-Lo, dark red and red sections, respectively). (E) i and iii median relative contribution of the 96 trinucleotide substitutions for $HM_{CpG-Hi}$ and $HM_{CpG-Lo}$ tumours, respectively, ii and iv Heatmap showing the cosine similarity score (CSS, white = 0, dark red = 1) of the mutational signatures in (i) and (iii) with the COSMIC signatures, respectively. Numbers denote SBS signatures from 1 (top left) to 94 (bottom right). For boxplot, the black central band represents the median. The lower and upper hinges represent the first and third quartiles, respectively. The whiskers represent the 1.5× interquartile range.

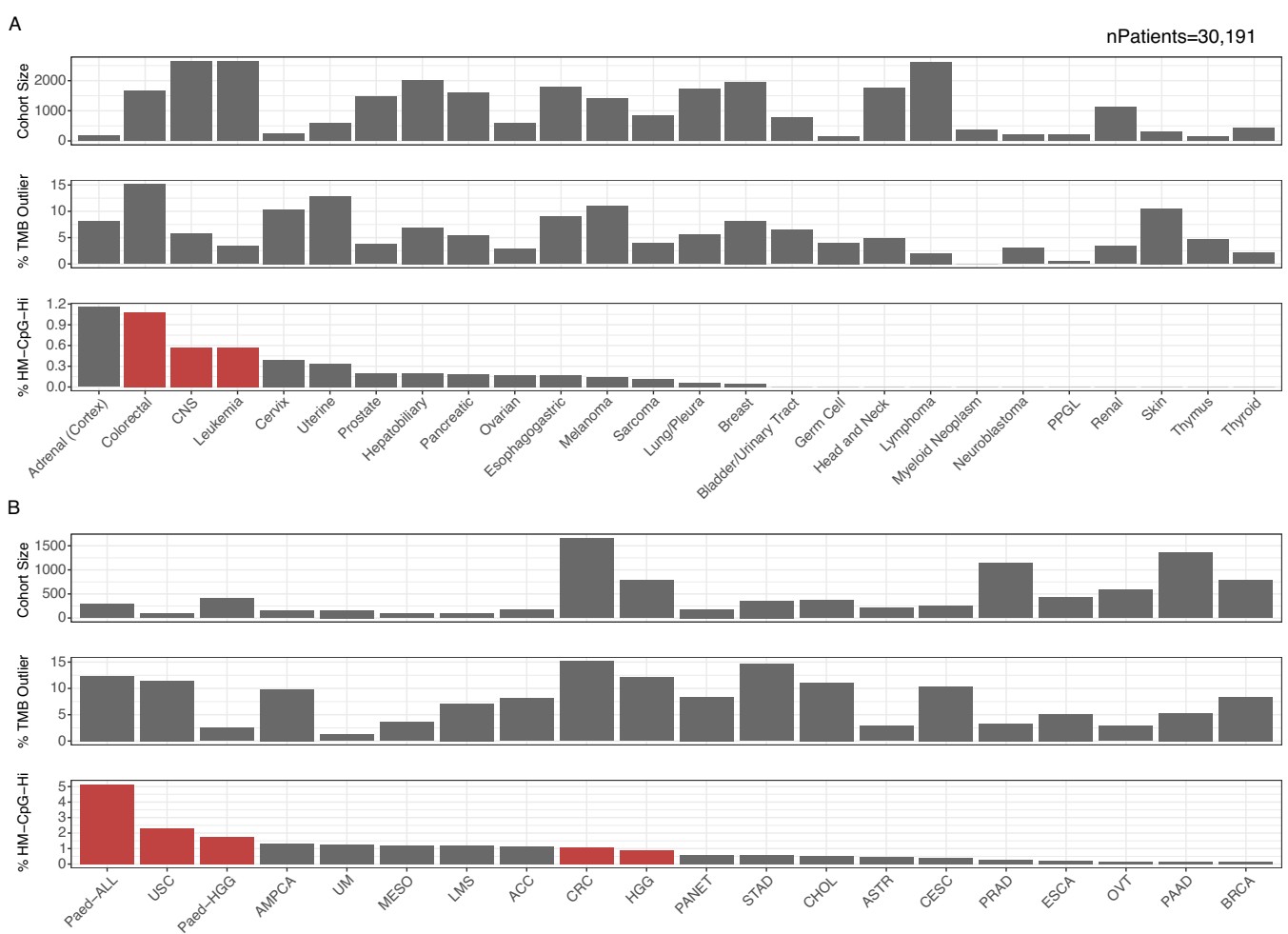

**Figure 2. Pan-cancer prevalence of the somatic CpG hypermutator phenotype.**

Summarised cohort size and mutation burden proportions across 26 tissue types (**A**) and 20 cancer types (**B**) representing 30,950 samples from 30,191 patients from 194 publicly available datasets. The number of tumour samples (top), the proportion of HM tumour samples (middle) and the percentage of CpG hypermutated tumour samples (bottom) per histology (**A**) and cancer type (**B**) reflects the same data seen in panel (**A**). However, the samples are grouped according to their detailed cancer-type annotation. Only cancer types exhibiting the somatic CpG hypermutator phenotype are shown. Enrichment in each cancer type was tested using a one-sided (greater-than) exact binomial test where the hypothesised probability of success was the frequency of CpG hypermutated samples across the whole dataset, the number of successes was the number of CpG hypermutated samples in the cohort, and the number of trials was the number of samples in the cohort (cohorts with $p$ values less than 0.05 are indicated with red colour) ($p$ values: Colorectal = 2.93e−7, CNS = 2.63e−3, Leukaemia = 2.88e−3, Paediatric-ALL = 1.73e−15, USC = 1.95e−2, Paediatric-HGG = 7.25e−5, CRC = 2.93e−7, HGG = 3.42e−3).

contributed by paediatric and adult high-grade gliomas (HGG) (1.2%, 14/1179) (Fig. 2B). While UV-mediated cutaneous melanoma is associated with a C > T mutagenesis signature distinct from CpG hypermutations, a small subset of the more rare uveal melanoma (UM) is associated with *MBD4*-associated CpG hypermutations (Johansson et al, 2020) (Fig. EV2). Beyond those cancer types traditionally associated with MMRd, we observed several cases of somatic CpG hypermutation in leukaemia (0.55%, 15/2719) (Fig. 2A). Despite considerable metastatic samples in the cohort ($n = 493$), the CpG hypermutator phenotype was predominantly observed in primary tumour specimens. While most of the cancer types associated with CpG hypermutation occurred in adults, paediatric cancers represented a significant enrichment (odds ratio = 5.2, $P = 3.4 \times 10^{-7}$, Fisher's exact test), in particular paediatric HGG (1.7%, 7/402) and paediatric leukaemia (5.1%,

15/291, Fig. 2B). This data demonstrates cancer type specific prevalences of somatic CpG hypermutation and a particular enrichment in colorectal cancer, paediatric HGG, and paediatric leukaemia.

## MutSα-deficiency associates with somatic CpG hypermutation in cancer

We next searched for gene alterations that might drive the somatic CpG hypermutation phenotype. The ratio of non-synonymous (dN) to synonymous (dS) mutations in a gene provides an established surrogate measure of selection during tumour evolution. We performed driver gene discovery for the entire cohort as well as independently for samples which had been identified as (CpG) hypermutators and restricted the analysis to previously established

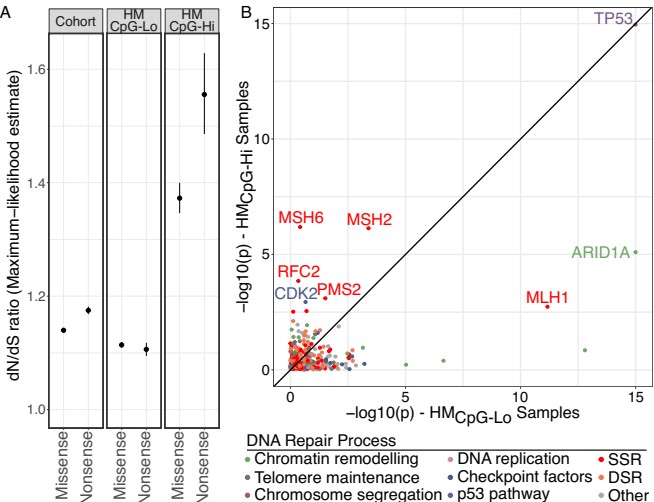

**Figure 3. Discovery of driver genes in cancers with somatic CpG hypermutation.**

(A) Quantification of selection during somatic tumour evolution (dN/dS ratio) for all patients ($n = 30{,}191$), somatic non-CpG hypermutators (HM$_{CpG-Lo}$, $n = 1914$), and somatic CpG hypermutators (HM$_{CpG-Hi}$, $n = 76$). Vertical lines denote a 95% confidence interval. (B) dN/dS-based driver gene discovery in tumours with somatic CpG hypermutation (y-axis) and other somatic non-CpG hypermutators (x-axis). Colours label genes in specific DNA repair pathways. See also Fig. EV1. $P$ values are based on likelihood-ratio tests.

cancer driver genes (Rheinbay et al, 2020) and DNA damage response genes (Pearl et al, 2015). Positive selection in cancer can be inferred by computing the dN/dS ratio while accounting for sequence context and tumour mutation burden. As expected, we found known driver genes to be associated with positive selection (dN/dS>1) for missense and nonsense mutations in the entire cohort and hypermutators (Fig. 3A). When exclusively focusing on somatic CpG hypermutators, we found an elevated dN/dS ratio of 1.37 for missense and 1.56 for nonsense mutations. This suggests that CpG hypermutators have a higher proportion of non-synonymous mutations compared to tumours with other forms of hypermutation and by inference, stronger signs of positive selection.

We next analysed selection coefficients (dN/dS) for individual driver genes. Among a list of known driver genes, somatic mutations in the *TP53* gene displayed the most significant signs of positive selection across all hypermutators ($q < 1e{-}16$) (Fig. 3B; Table EV3). Interestingly, we found a strong association between somatic CpG hypermutation and alterations in both MMR genes *MSH2* and *MSH6* ($q = 4.77e{-}4$ and $q = 1.43e{-}5$, respectively). In contrast, both genes showed no signs of positive selection in tumours with non-CpG hypermutator phenotypes ($q = 0.15$ and $q = 0.87$, respectively). Moreover, we found 0.57% of TMB-Hi tumours to have *POLE* hotspot mutation, but none in CpG hypermutated tumours. However, mutations in the MutSα complex (*MSH2/MSH6*) were enriched in CpG-HM compared to both TMB-high (OR = 2.5, $P = 0.001$) and TMB-low tumours (OR = 48, $P < 2.2e{-}16$). MSH2 and MSH6 form the MutSα heterodimer that is associated with detecting mismatches and initiating mismatch repair during replication. In contrast, somatic mutations in *MLH1*, part of the MutLα complex component, showed significant signs of positive selection in tumours with other forms of somatic

hypermutation (Fig. 3B). The DNA glycosylase *MBD4* has been previously associated with somatic CpG hypermutation in AML and uveal melanoma (Sanders et al, 2018; The ICGC/TCGA Pan-Cancer Analysis of Whole Genomes Consortium, 2020); however, all patients developed tumours due to genetic predisposition. Our driver analysis relied almost exclusively on somatic mutations and could not detect germline *MBD4* mutations as drivers of somatic CpG hypermutation. These results suggest that somatic *MBD4* mutations are rare in tumours with CpG hypermutation.

To explore other mechanisms of somatic CpG hypermutation, we examined available germline (28/76), somatic mutations, and gene expression in all tumours with a CpG hypermutator phenotype (Figs. 4A–F and EV1). We grouped tumour samples into those with a germline mutation, biallelic inactivation, monoallelic inactivation, low gene expression, and no alteration. Out of the ascertainable samples, we again found the MutSα complex genes to be the most recurrently mutated genes in CpG hypermutators, with 43% (32/76) of samples demonstrating a deleterious alteration in DNA repair genes (Fig. 4C). Additionally, we noted frequent low expression in the absence of *MSH2* mutations (3/76, 4%), suggestive of previously reported epigenetic silencing (Herman and Baylin, 2003). We found pathogenic germline mutations in *MSH2* (1/29), *MSH6* (1/29), and *MBD4* (3/29) and tumour loss of heterozygosity in 80% (4/5) of affected cases. Moreover, we noted many HGG patients with germline mutations in MMR genes (7/15 tumours had germline information available). In 50% (3/6) of HGG patients and the paediatric HGG patient (1/1) we discovered a pathogenic germline MMR gene variant. Somatic CpG hypermutation and biallelic inactivation were observed for *MSH2* (5/76 tumours) and *MSH6* (11/76 tumours), supporting a strong genetic association between MutSα-deficiency and somatic CpG hypermutation across cancer types.

## Somatic CpG hypermutation is exclusively present at relapse in paediatric leukaemia

Given our findings of somatic CpG hypermutation in many paediatric cancers (Fig. 2), we examined the occurrence during tumour evolution (diagnosis vs relapse) and across three cohorts consisting of 124 paired paediatric ALL (pALL) patients (Li et al, 2020; Ma et al, 2015; Rokita et al, 2019). We found CpG hypermutation in 6% (14/248) of pALLs; however, this was exclusively seen at relapse (11%, 14/124 vs 0%, 0/124 at diagnosis) (Fig. EV3C, E) ($p = 8.3e{-}5$). Therefore, we examined whether the CpG hypermutator phenotype was associated with one of the previously defined mutational signatures to understand better whether the observed mutation spectrum resembles an endogenous mutational process or one of the several treatment-associated signatures (Alexandrov et al, 2013). In agreement with our pan-cancer analysis (Fig. 1), relapse pALL showed almost exclusive association with the endogenous mutational signature SBS1 (present in all relapse samples, with 11/14 of the samples being exclusively attributable to SBS1 out of ascertainable SBS signatures, Fig. EV3D). The relapse-specific CpG hypermutator phenotype was associated with concordant somatic mutations in MMR genes in 79% (11/14) pALL patients. Interestingly, biallelic inactivation of *PMS2* was seen exclusively in two pALL cases with a CpG hypermutator phenotype, and both events occurred only in the relapse setting.

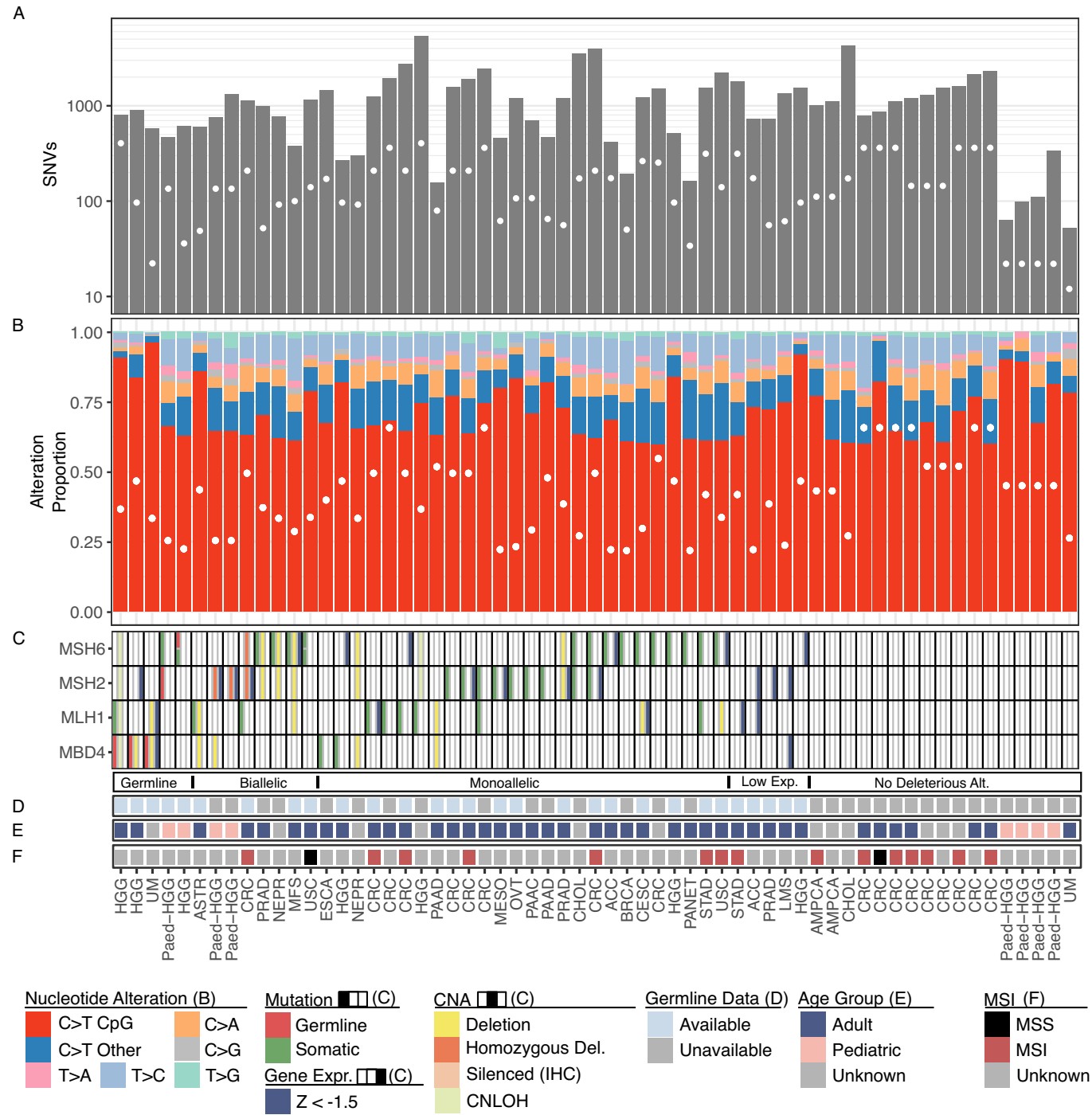

**Figure 4. MMR deficiency is associated with somatic CpG hypermutation in cancer.**

(A) The number of single nucleotide variants found in each specimen on a log10 scale, and the 75th percentile for the respective cancer type is indicated by a white dot. The relative proportion of each of the six transition and transversion events was calculated for each sample. (B) The proportion of somatic C > T mutations was further divided into those occurring in an NpCpG sequence context and in other contexts. A white dot marks the 75th percentile of the proportion of C > T mutations at CpG sites for the respective cancer type. (C) The presence of a genetic lesion (left column), copy number alteration (middle column), or relative reduction in z-scaled gene expression values (right column) in members of the base excision and DNA repair pathways. Information about germline data availability is shown in (D), the age group in (E) and microsatellite instability status in (F).

## Somatic CpG mutagenesis converges on *TP53*-deficiency

The most frequent somatic point mutations in *TP53* result in C > T mutations at CpG sites, representing about 25% of all *TP53* mutations in cancer (Olivier et al, 2010; IARC TP53 Database). This is notable since our driver analyses demonstrated that *TP53* mutations are selected during the evolution of tumours with CpG hypermutation. We thus hypothesised that somatic CpG hypermutation might accelerate tumour evolution due to failure to maintain methylated CpG sites in *TP53*. To this end, we analysed the sequence context of 39,925 somatic *TP53* mutations from the GENIE database (R9, Jan 2021) (Bouaoun et al, 2016). In agreement with previous studies, we found an 8–17-fold enrichment of somatic C > T mutations at CpG dinucleotides in *TP53* (Fig. EV4A). In total, 33% of all *TP53* mutations and 82% of recurrent hotspot sites—defined as mutation sites representing 1% or more of all *TP53* mutations—are found at CpG dinucleotides (GENIE database, Tables EV4, EV5). We next investigated whether tumours with a CpG hypermutator phenotype tended to have higher rates of somatic *TP53* mutations that occur at CpG sites. For non-CpG hypermutated tumours, we observed that 25% (203/806) of all somatic non-synonymous *TP53* mutations were C > T mutations at CpG sites, consistent with the prior observed distribution (GENIE database). In contrast, tumours with a CpG hypermutator phenotype exhibited a three-fold higher rate of somatic C > T mutations at CpG sites (75%, 33/44) (Fig. EV4B; Table EV4).

## Somatic CpG hypermutation is associated with response to checkpoint inhibitor therapy

MMRd and microsatellite Instability (MSI) are approved biomarkers for immune checkpoint inhibitors ICI treatment of advanced or recurrent solid tumours (Food et al, 2018), and it has been proposed that *MSH2/MSH6* mutations can also be used as a biomarker for predicting response to ICI (Sahin et al, 2019). Patients with a TMB of at least ten coding mutations per megabase (TMB-Hi) are approved by the FDA for ICI treatment. Using this established TMB cutoff, we investigated whether our CpG hypermutator threshold would enable further stratification of patients. To this end, we leveraged a large clinico-genomic cohort of 1661 ICI-treated patients with advanced cancer (Samstein et al, 2019) and separated patients into tumours with high TMB (using the FDA-approved cut-off of ≥10 coding mutations/Mb), CpG hypermutators (≥10 coding mutations/Mb and CpG>TpG mutation rate >0.60), and low TMB (<10 coding mutations/Mb). We identified a somatic CpG hypermutator phenotype in 1.1% (19/1661) of tumours (Fig. 5A; Table EV5), and a strong enrichment of mutations in the MutSα complex genes (*MSH2/MSH6*) in tumours with a somatic CpG hypermutation phenotype (53%) relative to tumours with a high (12%, OR = 8.5, P = 2.7e−5) and low TMB (0.9%, OR = 113, P = 2e−14) (Fig. 5B). Moreover, somatic CpG hypermutation accounted for a high proportion of colorectal cancers, bladder cancers, esophagogastric cancers, and head & neck cancers, and gliomas (Fig. 5C). Finally, we assessed whether somatic CpG hypermutation has predictive relevance for ICI therapy (PD-1/PDL-1, CTLA4, or both). Unadjusted and adjusted Cox proportional hazards regression models demonstrated improved clinical outcomes of ICI-treated patients that presented with either a CpG hypermutation phenotype (HR 0.27, 95%-CI

0.07–0.70, P = 0.01) or high TMB (HR 0.58, 95%-confidence interval [CI] 0.49–0.70, P < 0.001) (Figs. 5D and EV5A). We also found patients with a CpG hypermutator phenotype to be associated with improved survival when compared to patients with TMB-Hi (HR = 0.33, 95%-CI = 0.1–1.1, P = 0.07, Fig. EV5B). We note that somatic CpG hypermutators exhibit a particularly high TMB (42.3 mutations/Mb, Fig. 5E) and, in line with our discovery cohort, a high prevalence of *MSH2/MSH6* mutations, two previously described predictive biomarkers of ICI therapy response that might explain our observed association between ICI therapy response and CpG hypermutation status. Together, these results validate our pan-cancer observations that CpG hypermutation has the highest prevalence in colorectal cancer and glioma, is genetically linked to *MSH2/MSH6*-deficiency, and highlights a potential predictive relevance for ICI therapies.

## Discussion

Recently, interest in understanding the origin of somatic hypermutation has increased due to the clinical utility of tumour mutation burden (TMB) as a biomarker for ICI (Le et al, 2015). Using an objective criterion to discover somatic CpG hypermutation in cancer, we identified that 3.4% of all tumours with a hypermutation phenotype display somatic CpG hypermutation. Our results revealed that somatic CpG hypermutation is seen in 1% of colorectal cancer patients and expanded our knowledge about somatic hypermutation phenotypes in gastrointestinal tumours. A high prevalence in colorectal cancer patients is noteworthy considering the recent addition of colorectal adenomas to the tumour spectrum of individuals with very rare biallelic loss-of-function germline variants in *MBD4* and a somatic CpG hypermutation phenotype (Palles et al, 2022). We further found a high prevalence of somatic CpG hypermutation in paediatric HGG (1.7%) and paediatric ALL (5.1%). Previous studies described CpG hypermutation in *MBD4*-deficient early-onset AML (Sanders et al, 2018), and our study further expands it to paediatric ALL and paediatric-type diffuse high-grade gliomas. We demonstrate that both somatic and germline mutations in the mismatch repair complex MutSα (*MSH2* and *MSH6*), and to a very limited extent MutLα (*PMS2* and *MLH1*), are strongly associated with somatic CpG hypermutation. *MLH1*-mutant tumours can also acquire a high proportion of C > T mutations at CpGs (Németh et al, 2020), although this was not linked to a hypermutation phenotype. *MLH1* is frequently silenced by promoter hypermethylation (Cancer Genome Atlas Network, 2012), which we corroborate in several tumours using gene expression (Fig. 4). We note that this is likely to be an underestimate of the true rate of *MLH1* promoter hypermethylation. Paediatric-ALL was the only tumour type with somatic, relapse-associated *PMS2* mutations. This pattern suggests that *PMS2*-mutated paediatric ALL are 'primed' to undergo therapy-related somatic CpG hypermutation.

We find that somatic CpG hypermutation accelerates a signature resembling the 'clock-like' mutational signature SBS1, often used to time the number of cell divisions (Alexandrov et al, 2015). Our findings may, therefore, have direct implications for the utility of this ubiquitous and endogenous mutational signature as a mutational 'clock' since tumours with *MBD4*, *MSH2* or *MSH6* mutations will lead to an overestimation of mutational age when

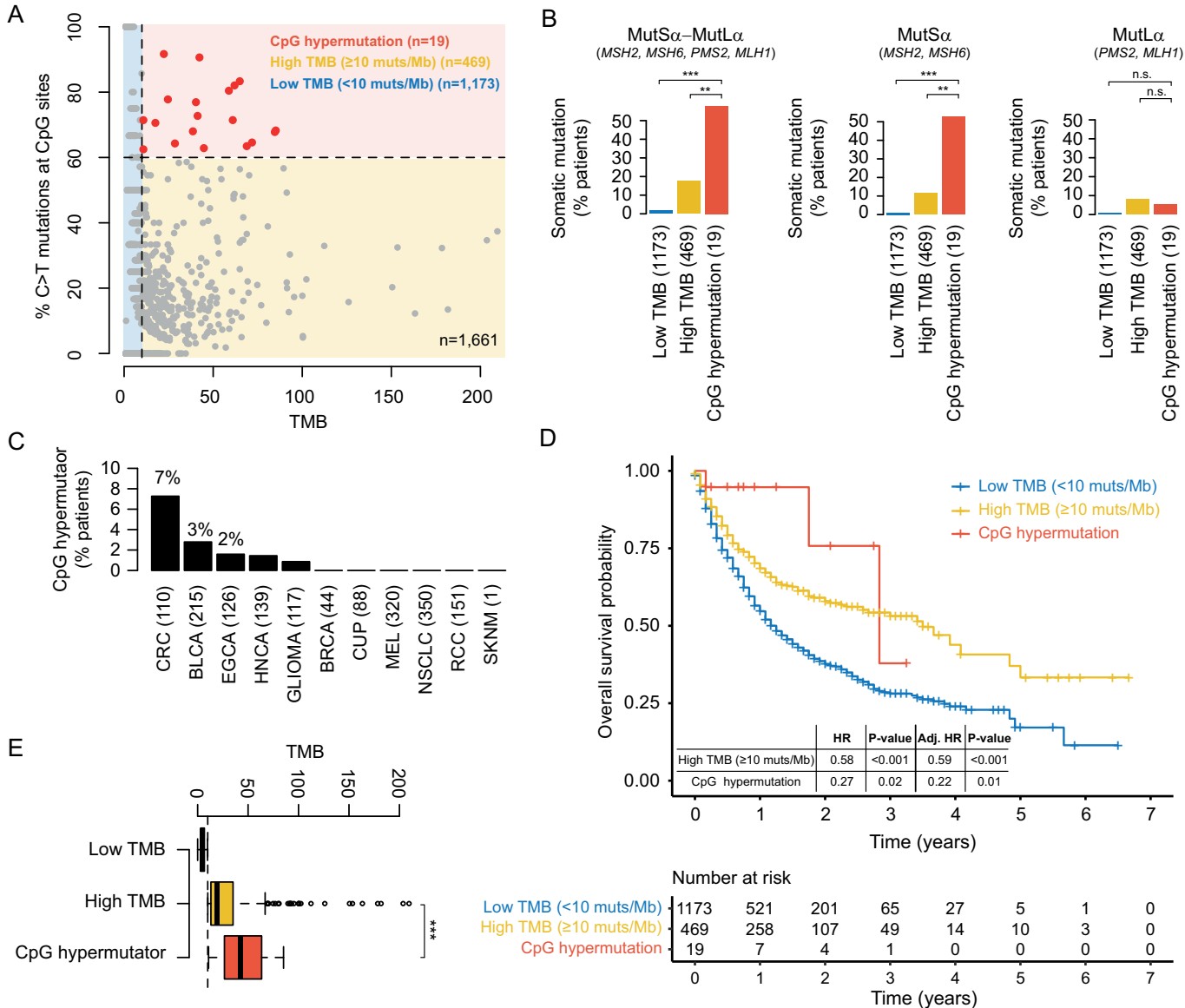

**Figure 5. Somatic CpG hypermutation and response to immune checkpoint inhibition.**

(A) Presence of somatic CpG hypermutation across 1,661 ICI-treated advanced cancer patients. (B) Enrichment of somatic MMR gene mutations in CpG-hypermutant tumours. Statistical significance was assessed using a Fisher's exact test (** denotes $P < 0.01$, *** denotes $P < 0.001$). MutSα-MutLα: CpG hypermutation vs TMB-low, $P = 3.4e-14$; CpG hypermutation vs TMB-high, $P = 1.5e-4$; MutSα: CpG hypermutation vs TMB-low, $P = 2.0e-14$; CpG hypermutation vs TMB-high, $P = 2.7e-5$; MutLα: CpG hypermutation vs TMB-low, $P = 0.15$; CpG hypermutation vs TMB-high, $P = 1.0$. (C) Frequency of CpG hypermutation across cancer types. (D) Kaplan–Meier overall survival curve of 1,661 ICI-treated patients stratified by low TMB, high TMB, and CpG hypermutation. Statistical significance was assessed using the Cox proportional hazards regression model. (E) Elevated TMB in CpG-hypermutant tumours. TMB-low, $N = 1173$; TMB-high, $N = 469$; CpG hypermutator, $N = 19$; Statistical significance was assessed using the Mann–Whitney U test (***$P < 0.001$). CpG hypermutator vs TMB-high, $P = 4.4e-4$. For boxplot, the black central band represents the median. The lower and upper hinges represent the first and third quartiles, respectively. The whiskers represent the 1.5× interquartile range. ***, $P < 0.001$.

compared to BER/MMR-proficient tumours. This also suggests that MutSα-mutated tumours display the unrepaired and potentially complete spontaneous deamination rate of 5-methylcytosine and that the MMR machinery plays a pivotal role in the repair at these sites, potentially guided by *MBD4* for strand-discrimination in non-replicating dsDNA (Chen and Furano, 2015; Fang et al, 2021).

Deficiency in MMR has been linked with accelerated mutagenesis preferentially in early-replicating and gene-rich regions of the genome (Supek and Lehner, 2015). The MMR complex is recruited to chromatin via the interaction of MSH6 with SETD2 and the H3K36me3 histone modification (Li et al, 2013). The vast majority of the 28 million CpG dinucleotides in the human genome are methylated in normal cells, and methylated CpG sites are thought to primarily act by repressing nearby gene expression via modulation of transcription factor binding (Lea et al, 2018; Yin et al, 2017). Hence, MMR genes play a crucial function in repairing spontaneous 5mC deamination events and keeping the epigenome functional and dynamic.

Our results also suggest that MutSα-deficiency is linked with an accelerated somatic mutation rate at cancer driver hotspots within *TP53*. We hypothesise that CpG hypermutation may bias the somatic mutational landscape, which can cause the selection of somatic driver mutations at methylated CpGs and within key protein domains such as the *TP53* DNA binding domain. Our findings provide a potential explanation for why *TP53* mutations showed the most significant association with the CpG hypermutation phenotype, even though *TP53* is not directly linked with mismatch repair. Moreover, somatic mutations in *TP53* and *MSH2* have been shown to act synergistically to promote tumour development (Toft et al, 2002), potentially bypassing p53 checkpoints and allowing increased genomic instability. Our study provides a model of tumour evolution whereby *MSH2* or *MSH6*-deficiency triggers unrestricted mutagenesis at methylated CpGs during the pre-malignant phase of tumour evolution and that the highly sequence-specific CpG hypermutator phenotype converges on *TP53* hotspot driver mutations and malignant transformation or relapse due to *TP53*-deficiency.

Finally, we find that adult cancer patients with a somatic CpG hypermutator phenotype show an improved clinical outcome after immune checkpoint inhibition, also after accounting for the overall tumour mutation burden. A recent case study reported an exceptional response to ICI in a patient with uveal melanoma, biallelic loss of *MBD4*, and a somatic CpG hypermutation (Rodrigues et al, 2018). The improved response of CpG hypermutators to ICI may be due to an increased somatic mutation rate in CpG hypermutators and an increased proportion of non-synonymous mutations. In addition, we speculate that additional properties could be involved, attributed to the distribution of 5mCpG sites in the human genome. For example, removal of methylated CpGs (to TpGs) may cause a general de-repression of enhancers and thus dysregulation of gene expression and activation of endogenous retroviruses. This could activate the immune response by activating endogenous interferon-based immune responses and/or the expression of novel neoantigens. Combining epigenetic drugs and immunotherapy is gaining increasing momentum (Villanueva et al, 2020). For example, 5-aza (5-aza-2-deoxycytidine) has been shown to induce the expression of dsRNA, which can induce a viral mimicry that activates an immune response (Liu et al, 2018; Roulois et al, 2015). Also, it was previously demonstrated that different DNA repair deficiencies drive distinct mutational dynamics during tumour evolution with, for example, *POLE*-deficiency causing early bursts of hypermutation. In contrast, MMRd drives late bursts of hypermutation in gastrointestinal and CNS tumours (Campbell et al, 2017; Chalmers et al, 2017). This kinetic hypermutation model suggests that somatic CpG hypermutation might continuously generate neoantigens. In contrast, other sources of hypermutation (such as UV light) might produce neoantigens during an early phase of tumour evolution. While our findings point to the potential utility of CpG hypermutation phenotype in the stratification of patients for ICI therapy, the number of CpG hypermutator patients in the ICI-therapy cohort was modest, and we acknowledge that expanded and diverse clinical cohorts will be needed to validate and extend our findings.

Overall, our analysis of whole exomes and genomes from over 30,000 cancer patients and 100 cancer types revealed MMR-associated somatic CpG hypermutation as a distinct hypermutator phenotype, most prominently in colorectal cancer, paediatric high-grade glioma, and paediatric leukaemia. We also demonstrate that MMR-associated somatic CpG hypermutation is compatible with an acceleration of the ubiquitous clock-like mutational process SBS1 and that CpG-hypermutant tumours converged on *TP53*-deficiency. Finally, our results suggest that immune checkpoint inhibitors benefit patients with a somatic CpG hypermutation phenotype.

# Methods

## Datasets

Preprocessed somatic mutation data were obtained from a total of 459 independent studies through the cBioPortal DataHub and data portal, the GLASS Consortium Synapse repository, and additional publications (Ma et al, 2015; Li et al, 2020) (see also Table EV6). The datasets contained a combination of whole-genome, exome, and panel sequencing data aligned against either GRCh37 or GRCh38. All panel-based datasets and any samples marked as derived cell lines in their meta-data were excluded from further analysis. To homogenise the remaining data, the liftOver package for R was used to convert all coordinates to GRCh38. Where mutation consequence annotation was not supplied, missing information was amended using SNPeff v4.3r (Cingolani et al, 2012) for SNVs and InDels. Where available, gene-level copy number and RNA-seq gene-expression data were obtained from each source. The clinical annotation for histological type supplied with each dataset was manually reviewed to create a uniform label set across all datasets using the OncoTree classification system version oncotree_2020_02_01 (Kundra et al, 2021) (Table EV7). Patients with an annotated age of less than 18 years were marked as paediatric cases. Where an included patient was represented by more than one sample, a single sample was chosen to represent the patient in the outlier threshold determination. In cases where a primary and metastasis pair was present, the primary was used, when annotation was unavailable or multiple primary samples were present a sample was selected at random. When all samples available for a patient were annotated as metastases or relapse/recurrence then the patient was excluded ($n = 526$) from the outlier threshold determination but included in subsequent outlier detection. Any cohorts with less than 20 representative samples were excluded from the analysis, where a cohort is defined as specimens from a single study with a shared histological type and sequencing method.

## Identification of CpG hypermutator samples

Samples exhibiting both high total mutation burden and high proportional mutation burden at CpG sites were identified by outlier analysis using Tukey's rule (Fig. 1). First, we grouped the cancer samples by study, histological type, and sequencing method (WGS and WES). Next, the 75th percentile (Q3) and interquartile range (IQR) for the number of somatic mutations were computed for each group. Samples with 1.5 times the IQR greater than Q3 (Outlier-Threshold = Q3 + 1.5 (IQR)) were considered outliers. To compute the CpG mutation burden outliers, we first used SomaticSignatures (v2.22) (Gehring et al, 2015) to determine sequence contexts at somatic mutations. The 75th percentile and IQR for CpG-mutation-proportion were next calculated across

the whole cohort, and a cohort-wide outlier threshold was determined as above. CpG hypermutators were classified based on the following criteria, (i) mutation burden outlier (Tukey's method), (ii) C > T mutations at CpG-sites greater than the cohort-wide threshold (0.6), and (iii) tumour mutation burden greater than the median across all samples in the entire cohort (1.33 mutations/Mb). (the full code to compute CpG hypermutation is provided, see the Data availability section). The term $HM_{CpG-Hi}$ will be used throughout the manuscript to indicate tumours with a somatic CpG hypermutator phenotype. $HM_{CpG-Lo}$ will be used to describe a high mutation burden without enrichment of somatic mutations at CpG sites. To calculate a uniform tumour mutation burden, mutation calls were restricted to a minimal set of exome regions (https://bitbucket.org/cghub/cghub-capture-kit-info/src/master/BI/vendor/Agilent/whole_exome_agilent_1.1_refseq_plus_3_boosters.targetIntervals.bed). In the comparative analysis of CpG hypermutators to the FDA-approved TMB burden, CpG hypermutators were computed requiring ≥10 coding mutations/Mb and CpG>TpG mutation rate greater than 0.6. We applied P-MACD (Roberts et al, 2013) using default parameters for nCg and rCg models (The ICGC/TCGA Pan-Cancer Analysis of Whole Genomes Consortium, 2020) to obtain minimum estimates of mutation loads associated with an nCg-specific mutagenic mechanism (https://github.com/NIEHS/P-MACD). The P-MACD approach estimates the enrichment and mutation load of a suspected specific mutational process based on prior mechanistic knowledge about mutation motifs (Np[C > T]pG with N = A or C or T or G; Rp[C > T]pG with R = C or T) associated with somatic CpG mutagenesis. The nCg model includes all CpG sequence contexts for estimating CpG mutagenesis. The rCg model accounts for a subset of all CpG sites and was designed to exclude potential impacts of UV mutagenesis on our CpG hypermutator estimates given that the UV-associated mutational signature 7 (Alexandrov et al, 2013) includes two CpG sites (Cp[C > T]pG and Tp[C > T]pG). No blinding was performed in our analyses.

### Driver enrichment analysis

Identification of driver genes enriched in hypermutated samples was performed using the dNdScv package for the R statistical framework (Martincorena et al, 2017) using the default parameters. Mutation positions were converted to GRCh37 using the liftOver package before analysis. Somatic *TP53* mutations were downloaded from the International Agency for Research on Cancer (IARC) TP53 database (R20, July 2019). TP53 hotspots were defined as those representing 1% or more of the total population of mutations observed in the Genomics Evidence Neoplasia Information Exchange (GENIE) data in the IARC database.

### Survival analysis of response to ICI

We performed a survival analysis based on 1661 patients in the MSK-IMPACT immunotherapy study using Kaplan–Meier estimates and log-rank tests (Samstein et al, 2019). We used the FDA-approved cut-off of ≥10 mut/Mb, which includes all non-synonymous coding SNVs with a VAF >5% (Chalmers et al, 2017; Food et al, 2018) and separated patients into hypermutators (≥10 mutations/Mb, 'TMB-Hi') and non-hypermutators (<10 mutations/Mb, 'TMB-Lo'). Hypermutator tumours

were further separated into TMB-Hi$_{CpG-Hi}$ and TMB-Hi$_{CpG-Lo}$ cases based on the median proportion of somatic C > T mutations at NpCpG sites.

## Data availability

All data used in this publication was obtained from publicly accessible data sources (see Materials and methods). Data from the ICGC/TCGA Pan-Cancer Analysis of Whole Genomes Consortium may be subject to a data access control policy and require a data access agreement. The computer code for the identification of CpG-hypermutant tumours produced in this study is available in the following repository: - Computer Code: Bitbucket (https://bitbucket.org/weischenfeldt/cpghm_publication_code).

The source data of this paper are collected in the following database record: biostudies:S-SCDT-10_1038-S44320-024-00054-5.

## Peer review information

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

## Acknowledgements

JW was supported by the Danish Cancer Society (R147-A9843), the Danish Council for Independent Research (8020-00282) and the Novo Nordisk Foundation (NNF200C0060141). SMW was supported by the École Polytechnique Fédérale de Lausanne, the Research Council of Norway (187615), the South-Eastern Norway Regional Health Authority, and the University of Oslo. AJF was supported by the Danish Cancer Society (R147-A9843). The results here are in whole or part, based on data generated by the TCGA Research Network (www.cancer.gov/tcga) and the GLASS Consortium (www.glass-consortium.org). We thank Dr Balca R. Mardin for critical feedback on the manuscript.

## Author contributions

**Aidan Flynn**: Conceptualisation; Data curation; Software; Formal analysis; Investigation; Visualisation; Methodology; Writing—original draft; Writing— review and editing. **Sebastian M Waszak**: Conceptualisation; Supervision; Funding acquisition; Data curation; Software; Formal analysis; Investigation; Visualisation; Methodology; Project administration; Writing— review and editing. **Joachim Weischenfeldt**: Conceptualisation; Supervision; Funding acquisition; Visualisation; Methodology; Writing—original draft; Project administration; Writing—review and editing.

Source data underlying figure panels in this paper may have individual authorship assigned. Where available, figure panel/source data authorship is listed in the following database record: biostudies:S-SCDT-10_1038-S44320-024-00054-5.

## Disclosure and competing interests statement

The authors declare no competing interests.

# Expanded View Figures

**Figure EV1.   P-MACD CpG mutation load estimates in HM tumours and MMR gene mutations in CpG hypermutated tumours.**

(**A**) %nCg (with C > T) mutation load (min. estimate) derived for 1938 hypermutant (HM)-tumours using the P-MACD pipeline and stratified by tumours with somatic CpG hypermutator status (HM;CpGlo, $n = 76$ and HM;CpGhi, $n = 1860$). See Methods for more details about the nCg model. (**B**) nCg vs rCg ($r = $ A or G) min. mutation load estimates from P-MACD in CpG hypermutated tumours. See Methods for more details about the nCg and rCg model. (**C**) The distribution of mutation consequences for deleterious genomic alterations observed in core mismatch repair genes within CpG hypermutated samples. (**D**) Theoretical proportion of non-synonymous mutations (vertical axis) for each dinucleotide context (horizontal axis) based on codon usage for each amino acid, showing no bias towards non-synonymous mutation of codons in C>TpG context (orange colour). For boxplots, the black central band represents the median. The lower and upper hinges represent the first and third quartiles, respectively. The whiskers represent the 1.5× interquartile range.

                                                      

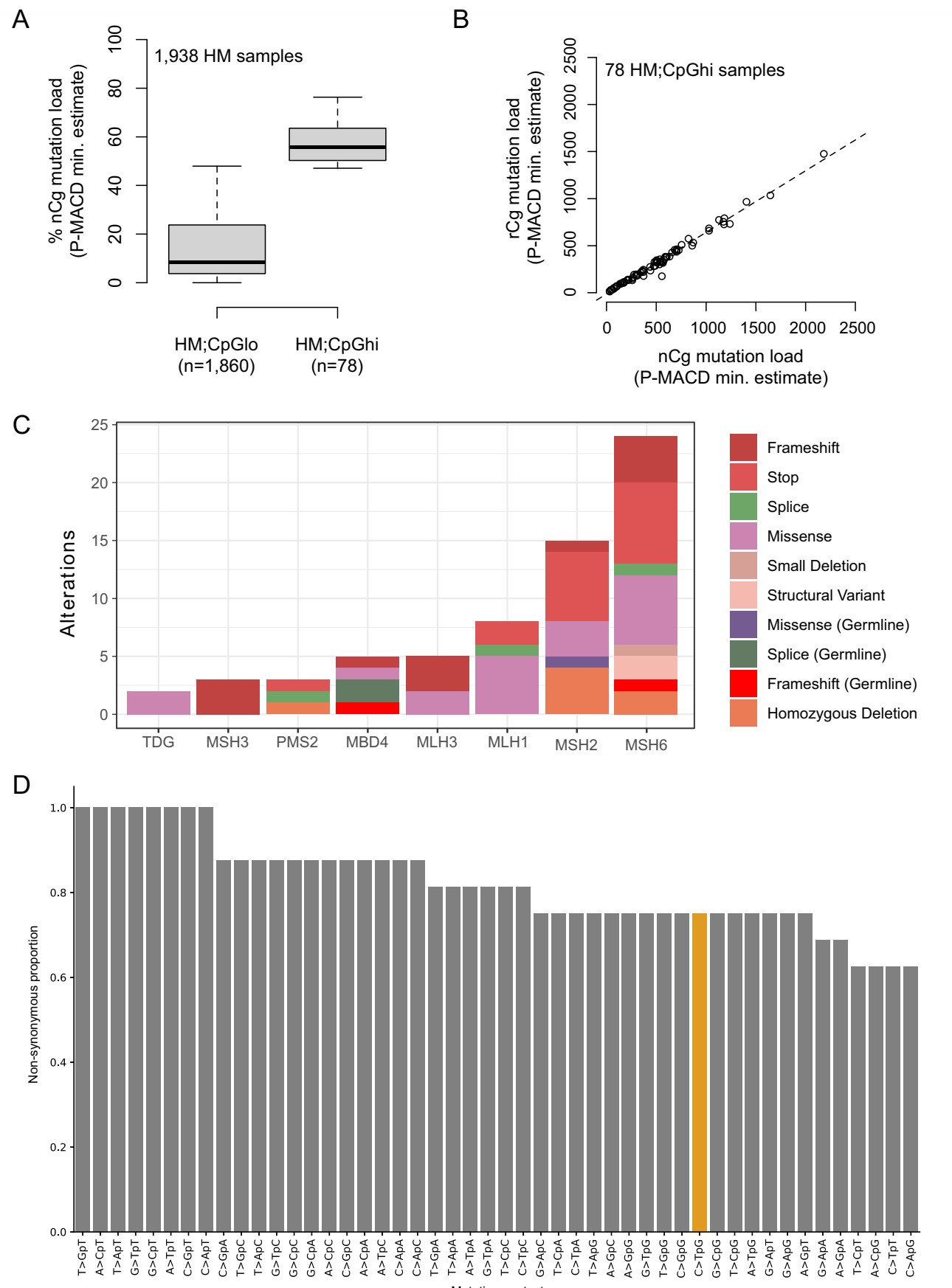

A

Uveal melanoma (uvm_tcga_pan_can_atlas_2018)

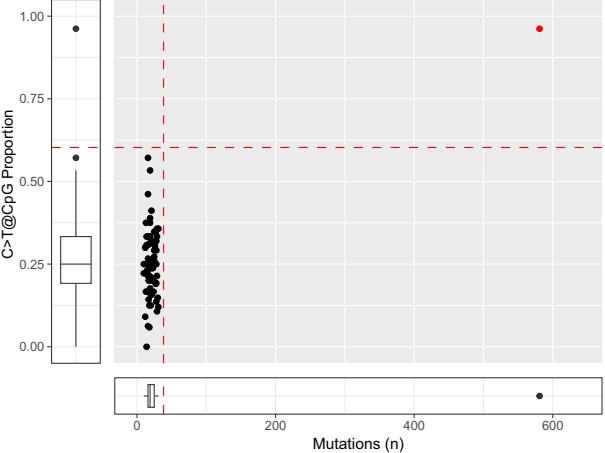

B

Skin cutaneous melanoma (skcm_tcga_pan_can_atlas_2018)

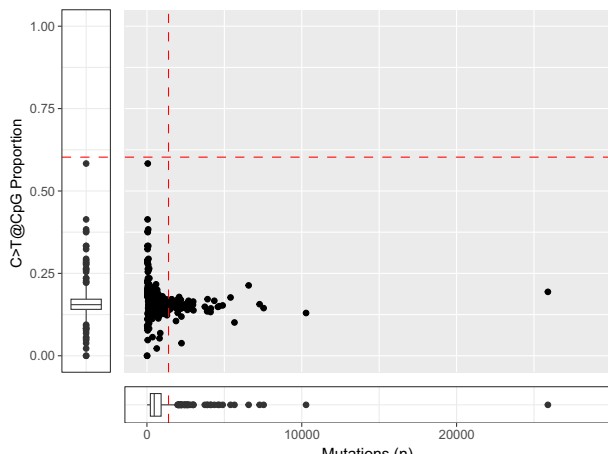

**Figure EV2.   Outlier analysis of skin and uveal melanoma.**

Outlier analysis was performed on the 2018 TCGA (**A**) skin cutaneous melanoma ($n = 406$) and (**B**) uveal melanoma ($n = 80$) cohorts by applying Tukey's fence to identify high mutation load tumours and the cohort-wide cutoff for C > T@CpG proportion (see Fig. 1). Tumours exceeding both thresholds were considered CpG-Hypermutators (red dots). For boxplots, the black central band represents the median. The lower and upper hinges represent the first and third quartiles, respectively. The whiskers represent the 1.5× interquartile range.

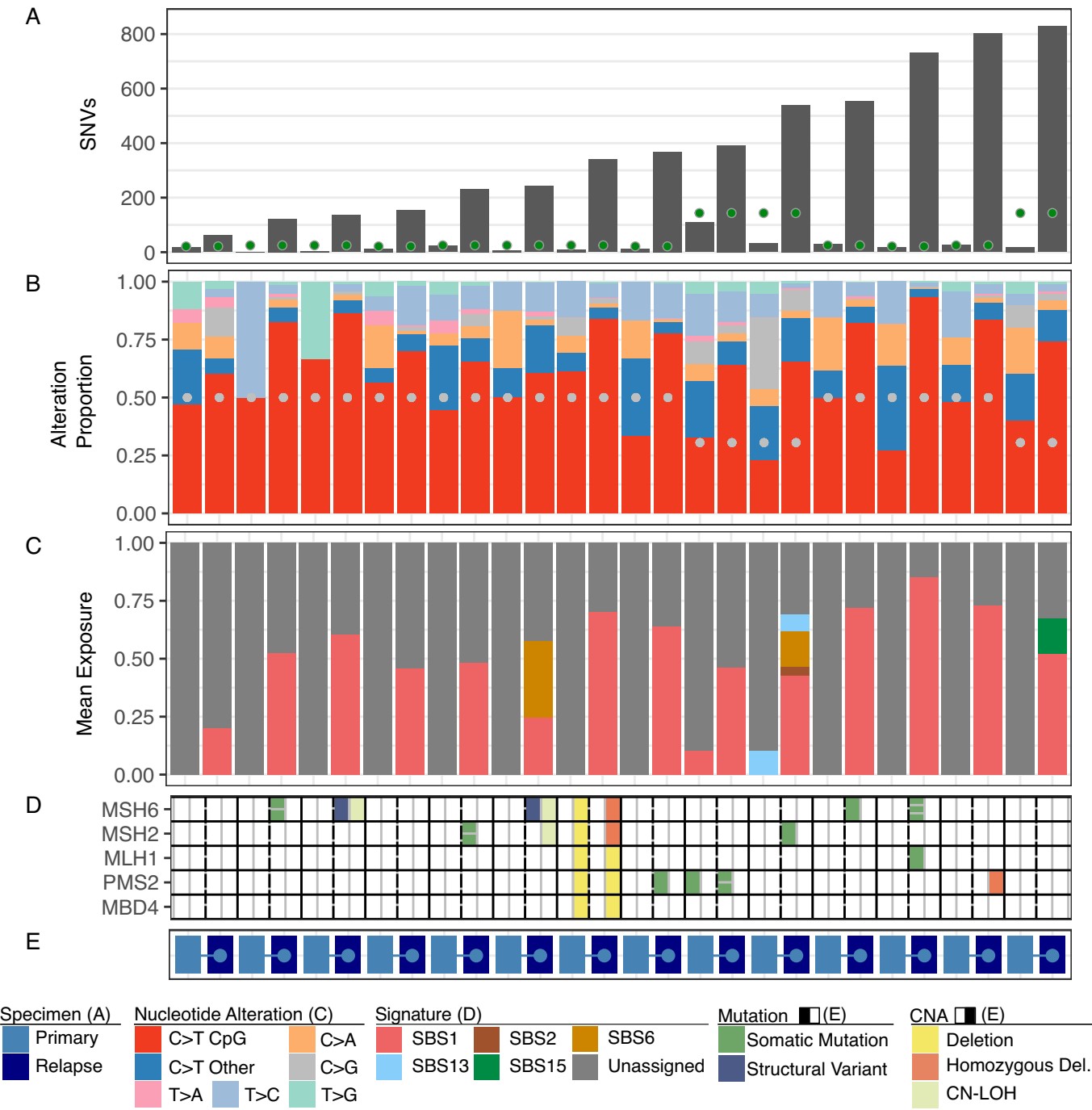

**Figure EV3. Relapse-specific somatic CpG hypermutation in paediatric ALL.**

Outlier analysis identified somatic CpG hypermutation in 14 paediatric acute lymphoblastic leukaemia. (A) The total number of somatic mutations found in each specimen. The 75th percentile for paediatric ALL in the cohort is marked by green dots. (B) The relative proportion of each of the six transition and transversion events, the proportion of C > T mutations were further divided into those occurring in a CpG context and in other contexts. Grey dots mark the 75th percentile for the proportion of mutations in a CpG context for paediatric ALL in the cohort. (C) Mean signature exposure was determined using the sigfit algorithm and COSMIC v3 mutational signatures. Signature assignment was not possible in many primary specimens due to low mutation load. (D) Genetic lesions (left column) and copy number alterations (right column) in members of the MMR pathway. (E) Mutation data were available from samples taken at the initial presentation (light blue), and after relapse (dark blue) Samples from a single patient are indicated by a linking line.

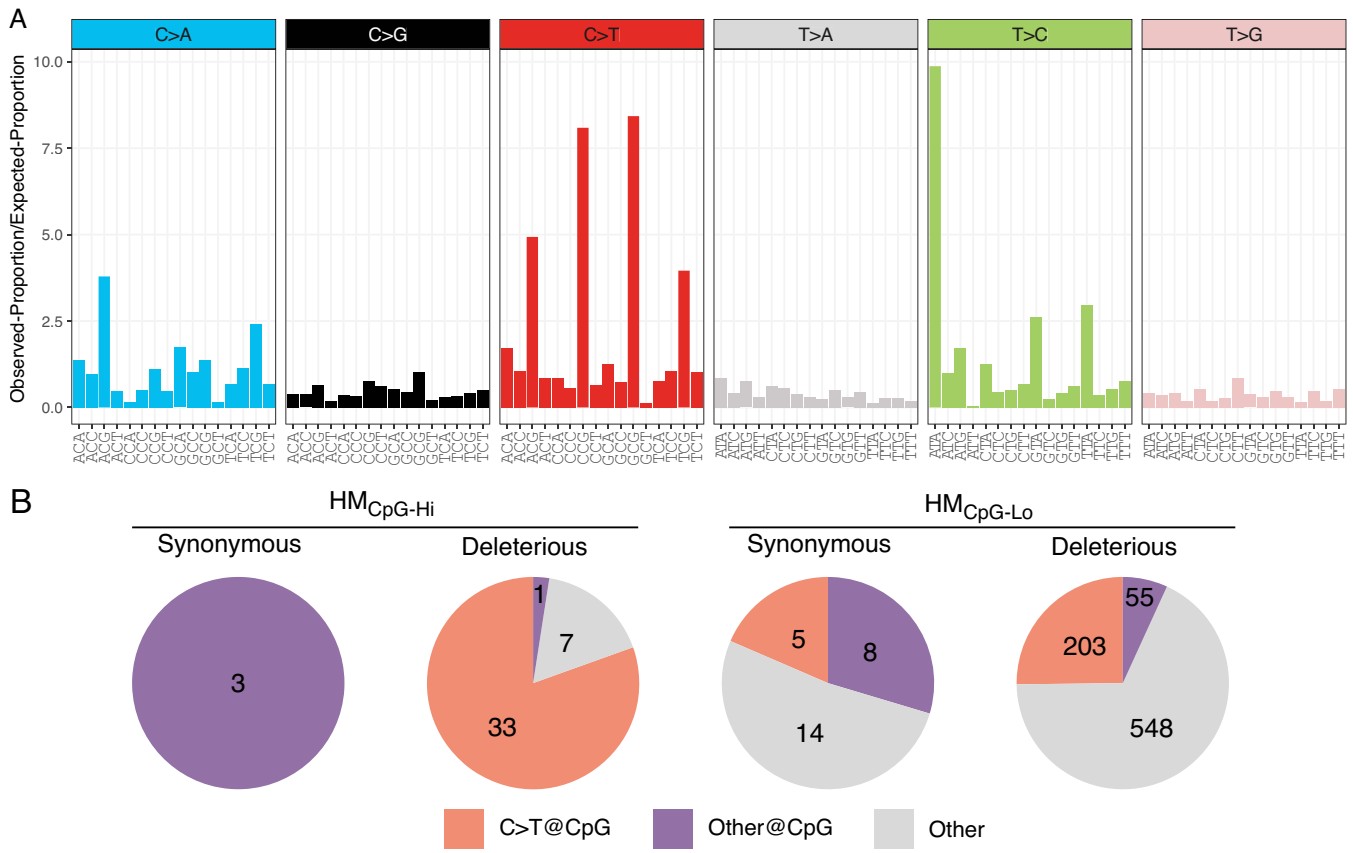

**Figure EV4. Somatic CpG hypermutation is associated with *TP53* hotspot mutations.**

(A) Observed/expected ratio of somatic mutations in trinucleotide contexts of the *TP53* gene. *TP53* mutation data were obtained from the GENIE database. The expected frequency for each mutational context was computed by determining the context for each mutation site and alternate-allele combination registered in the GENIE database, the sum of somatic mutations within each context was then expressed as a proportion of the total number of distinct TP53 mutations in the GENIE database ($n = 1467$). The observed frequency was computed as per the expected, however, each mutational site and the alternate-allele combination was counted once for each sample in which the mutation was observed in the GENIE database. The result was expressed as a proportion of the total number of *TP53* mutations observed in the GENIE database ($n = 39,925$). (B) Pie-chart of synonymous versus non-synonymous/deleterious SNVs in tumours with the CpG hypermutation phenotype and other somatic hypermutation phenotypes. Colours denote C > T mutations at CpG sites (orange), non-C > T mutations at CpG sites (purple) and all other mutations at non-CpG sites (grey).

 

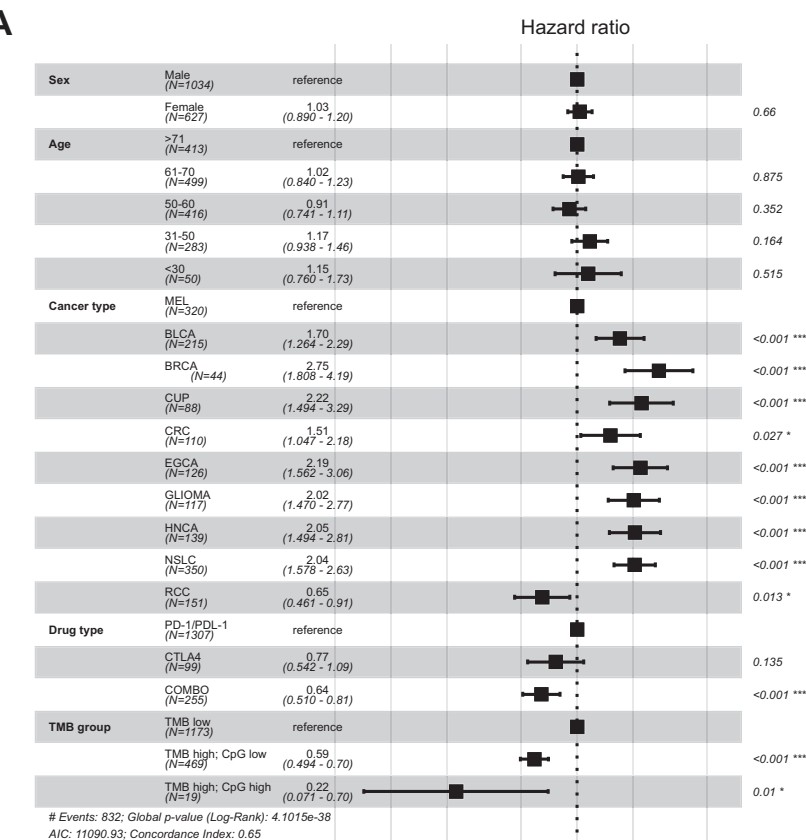

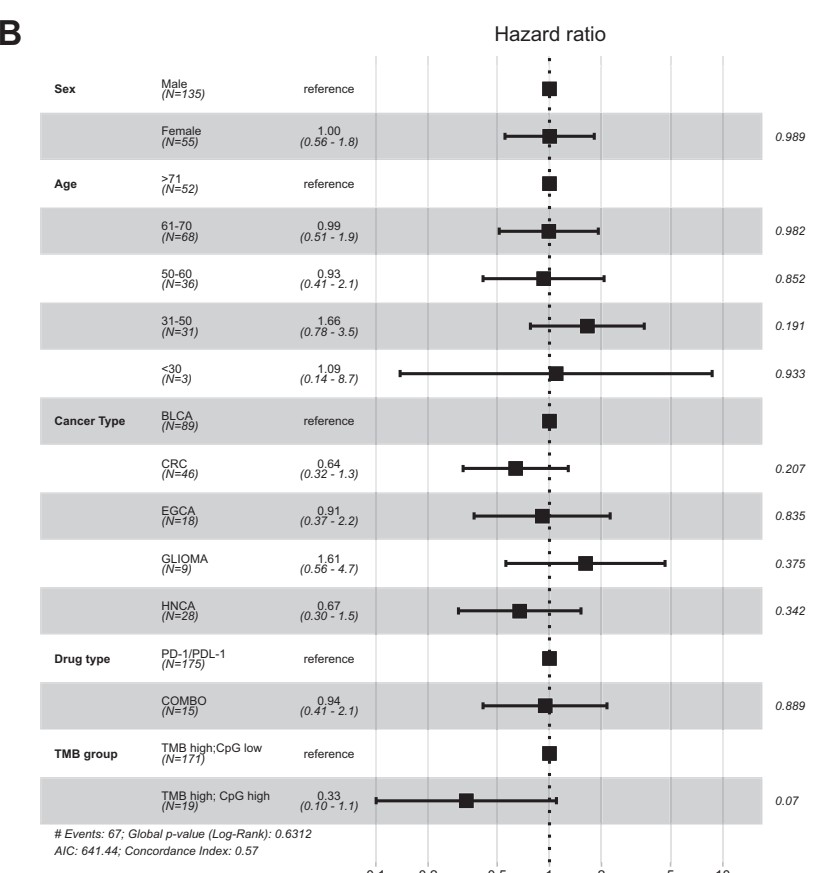

◀ **Figure EV5.   Forest plot for Cox proportional hazards model of overall survival in ICI-treated cancer patients.**

(**A**) Multivariate Cox proportional hazards model for overall survival for all patients. BLCA: $P = 4.5e{-}4$; BRCA: $P = 2.4e{-}6$; CUP: $P = 7.7e{-}5$; EGCA: $P = 5.3e{-}6$; GLIOMA: $P = 1.4e{-}5$; HNCA: $P = 8.6e{-}6$; NSLC: $P = 4.8e{-}8$; COMBO: $P = 1.7e{-}4$; TMB-high;CpG-low: $P = 2.7e{-}9$. Error bars, 95% confidence interval. (**B**) Multivariate Cox proportional hazards model for overall survival for TMB-high patients and cancer types with somatic CpG hypermutators. MEL melanoma, BLCA bladder cancer, BRCA breast cancer, CUP cancer of unknown primary, CRC colorectal cancer, EGCA esophagogastric cancer, GLIOMA glioma, HNCA head and neck cancer, NSCLC non-small cell lung cancer, RCC renal cell carcinoma, TMB tumour mutation burden. Error bars, 95% confidence interval.

