## [Peer Review File · Molecular Systems Biology]

Somatic CpG hypermutation is associated with mismatch repair deficiency in cancer

Joachim Weischenfeldt, Aidan Flynn, and Sebastian Waszak

Corresponding author(s): Joachim Weischenfeldt (joachim.weischenfeldt@bric.ku.dk) , Sebastian Waszak (sebastian.waszak@epfl.ch)

Review Timeline:

Submission Date:	7th Dec 23
Editorial Decision:	1st Feb 24
Revision Received:	12th May 24
Editorial Decision:	23rd May 24
Revision Received:	17th Jun 24
Accepted:	28th Jun 24

Editor: Poonam Bheda

Transaction Report:

1st Feb 2024

Manuscript Number: MSB-2023-12164-T

Title: Somatic CpG hypermutation is associated with mismatch repair deficiency in cancer

Author: Joachim Weischenfeldt

Sebastian Waszak

Aidan Flynn

Dear Prof. Weischenfeldt,

Thank you again for submitting your work to Molecular Systems Biology. We have now heard back from the three referees who agreed to evaluate your manuscript. As you will see below, the reviewers appreciate the findings presented in your manuscript. However, they raise a series of concerns, which we would ask you to address in a major revision.

I think that the recommendations of the reviewers are rather clear and I therefore do not see the need to repeat the comments listed below. All issues raised would need to be satisfactorily addressed. Please let me know in case you would like to discuss in further detail, I would be happy to schedule a call.

We require:

4) A .docx formatted letter INCLUDING the reviewers' reports and your detailed point-by-point responses to their comments. As part of the EMBO Press transparent editorial process, the point-by-point response is part of the Review Process File (RPF), which will be published alongside your paper.

5) A complete author checklist, which you can download from our author guidelines (<https://www.embopress.org/page/journal/17574684/authorguide#submissionofrevisions>). Please insert information in the checklist that is also reflected in the manuscript. The completed author checklist will also be part of the RPF.

6) Please note that all corresponding authors are required to supply an ORCID ID for their name upon submission of a revised manuscript.

7) It is mandatory to include a 'Data Availability' section after the Materials and Methods. Before submitting your revision, primary datasets produced in this study need to be deposited in an appropriate public database, and the accession numbers and database listed under 'Data Availability'. Please remember to provide a reviewer password if the datasets are not yet public (see <https://www.embopress.org/page/journal/17574684/authorguide#dataavailability>).

In case you have no data that requires deposition in a public database, please state so in this section. Note that the Data Availability Section is restricted to new primary data that are part of this study. This study includes no data deposited in external repositories.

8) For data quantification: please specify the name of the statistical test used to generate error bars and P values, the number (n) of independent experiments (specify technical or biological replicates) underlying each data point and the test used to calculate p-values in each figure legend. The figure legends should contain a basic description of n, P and the test applied. Graphs must include a description of the bars and the error bars (s.d., s.e.m.). Please provide exact p values.

9) Our journal encourages inclusion of *data citations in the reference list* to directly cite datasets that were re-used and obtained from public databases. Data citations in the article text are distinct from normal bibliographical citations and should directly link to the database records from which the data can be accessed. In the main text, data citations are formatted as

follows: "Data ref: Smith et al, 2001" or "Data ref: NCBI Sequence Read Archive PRJNA342805, 2017". In the Reference list, data citations must be labeled with "[DATASET]". A data reference must provide the database name, accession number/identifiers and a resolvable link to the landing page from which the data can be accessed at the end of the reference. Further instructions are available at .

<https://www.embopress.org/page/journal/17574684/authorguide#expandedview>

11) For more information: There is space at the end of each article to list relevant web links for further consultation by our readers. Could you identify some relevant ones and provide such information as well? Some examples are patient associations, relevant databases, OMIM/proteins/genes links, author's websites, etc...

12) Author contributions: CRediT has replaced the traditional author contributions section because it offers a systematic machine readable author contributions format that allows for more effective research assessment. Please remove the Authors Contributions from the manuscript and use the free text boxes beneath each contributing author's name in our system to add specific details on the author's contribution. More information is available in our guide to authors.

13) Disclosure statement and competing interests: We updated our journal's competing interests policy in January 2022 and request authors to consider both actual and perceived competing interests. Please review the policy <https://www.embopress.org/competing-interests> and update your competing interests if necessary.

14) Every published paper now includes a 'Synopsis' to further enhance discoverability. Synopses are displayed on the journal webpage and are freely accessible to all readers. They include a short stand first (maximum of 300 characters, including space) as well as 2-5 one-sentences bullet points that summarizes the paper. Please write the bullet points to summarize the key NEW findings. They should be designed to be complementary to the abstract - i.e. not repeat the same text. We encourage inclusion of key acronyms and quantitative information (maximum of 30 words / bullet point). Please use the passive voice. Please attach these in a separate file or send them by email, we will incorporate them accordingly.

Please also suggest a striking image or visual abstract to illustrate your article as a PNG file 550 px wide x 300-600 px high. Share synopsis text and image, as well as eTOC:

Please note that these would be the final versions and changes during proofing are usually not allowed

15) As part of the EMBO Publications transparent editorial process initiative (see our Editorial at <http://embomolmed.embopress.org/content/2/9/329>), Molecular Systems Biology Medicine will publish online a Review Process File (RPF) to accompany accepted manuscripts.

In the event of acceptance, this file will be published in conjunction with your paper and will include the anonymous referee reports, your point-by-point response and all pertinent correspondence relating to the manuscript. Let us know whether you agree with the publication of the RPF and as here, if you want to remove or not any figures from it prior to publication.

Molecular Systems Biology has a "scooping protection" policy, whereby similar findings that are published by others during review or revision are not a criterion for rejection. Should you decide to submit a revised version, I do ask that you get in touch after three months if you have not completed it, to update us on the status.

I look forward to receiving your revised manuscript.

Yours sincerely,

Poonam Bheda, PhD
Scientific Editor
Molecular Systems Biology

Reviewer #1:

The manuscript describes analysis of somatic mutation load associated with methylated cytosines in CpG sequences (CpG TpG). This is the well-known type of motif-specific somatic mutagenesis ubiquitous in normal and in cancer tissues. Authors analyzed two cohorts: (i) a vast dataset of over 30,000 cancer samples whole-exome (WES) or whole-genome (WGS) sequenced by many different projects and belonging to multiple tumor types; (ii) and a cohort of 1,661 tumor samples in which a panel of exomes and selected introns of 341 genes were captured and sequenced (the latter cohort described in ref. (Samstein et al, 2019). There are several more conclusions of interest to cancer field.

Main conclusions of the manuscript are as follows:

A. Based on analysis of mutation calls in the 30,000-sample WES+WGS cohort authors conclude that the somatic defects in MutS alfa (MSH2-MSH6 genes) are the strongest drivers of high level of CpG mutagenesis.

B. Based on analysis of 1,661-sample cohort in which only a panel of genes (mainly exons) was sequenced authors conclude that CpG mutagenesis is a better predictor of positive response to treatment with immune checkpoint inhibitors (ICI) than overall increase in tumor mutation burden (TMB).

This is interesting study which however requires correction of a significant inaccuracy as well as addressing other comments below.

1. Content of four Supplementary Tables presented as tabs of a single excel file

"MSB-2023-12164-T-Supplementary_Tables-supp.xlsx"

does not match descriptions in the text. Moreover, the text mentions six Supplementary Tables (Table S1 - Table S6). I suspect that the wrong excel file could have been uploaded at submission.

2. Page 8, lines 22-24

Quote: "The CpG hypermutated tumours, on the other hand, displayed a strong similarity to SBS1 (cosine 0.95)"

Comment:

In fact, cosine similarity to SBS1 for CpG-Hi (hypermutated) tumours as shown on Fig.1E(ii) is 0.29. The values on entire panel Fig.1E(ii), which is supposed to show cosine similarities for CpG-Hi (hypermutated) tumours, are exactly the same as on panel Fig.1E(iv), which is supposed to show cosine similarities for CpG-Lo (non-CpG hypermutated) tumours. The only difference is that panel iv is shaded in gray.

Is it some kind of mixup?

3, Page 6,

Line 7 from bottom

"using default parameters for nCg and rCg models"

The rationale for the use of these two models is never explained either in the text or in the legend of Supplementary Figure 1B

line 5 from bottom

"to estimate minimum mutation loads" can be replaced for "to obtain minimum estimates of mutation loads associated with nCg-specific mutagenic mechanism" to stress that the estimate is not just a simple mutation count.

4. Figure 1 and associated analysis.

Mutation Np[C>T]pG motifs represent significant fraction of total mutations in melanomas, which are also often contain very high overall mutation load. However, high level of Np[C>T]pG mutagenesis in melanomas can be ascribed to UV-mutagenesis - C>T in dipyrindines, TC>TT or CC>CT. Interestingly, only uveal melanomas were identified as CpG-Hi in this study. This should be explained with details of analysis in all 30000 plus samples, which is not present in the current Supplementary Table set.

5. Page 14, last para.

Quote:

"We find that somatic CpG hypermutation accelerates a signature resembling the 'clock-like' mutational signature SBS1, often used to time the number of cell divisions (Alexandrov et al, 2015), which suggests a replication-dependent MMR pathway, involving error-prone processing of MutS α ."

Comment:

In fact, (Alexandrov et al, 2015) explain higher presence of SBS1 in tumors from tissues with high cell division rates by smaller times of T:G mismatches existence between rounds of replication, where a C>T change would be copied by DNA Pol and fixed into mutation. Only at the stretch of time a T:G mismatch exists in double-strand DNA, a thymine in the mismatch can be repaired by templated repair mechanism - BER or MMR. Importantly, in case of MMR, an unknown mechanism of strand-discrimination in non-replicating double-strand DNA should be postulated. Fang et al 2021 cited in this submission propose that error-free mismatch repair or BER of T:C mismatches created by cytidine deamination meCpG is aided by MBD4. In case of

MMR, meC-specific glycosylase MBD4 can provide a tool for strand-discrimination by MMR.

6. Supplemental Figure 1.

Panel B: Explain the reason of looking at rCg vs nCg.

Panel C. POLE and POLD1 are catalytic large subunits of replicative DNA polymerases Pol-epsilon and Pol-delta, respectively. Their roles in MMR are not well defined. TDG and MBD4 are components of BER. Only MBD4 was shown to interact with MMR and was proposed to provide strand-discrimination for non-canonical MMR (see comment to page 14 above) This should be indicated.

Reviewer #2:

This work by Flynn et al. represents a large scale analysis of existing cancer sequencing data focusing on somatic CpG hypermutation. The analysis suggests that somatic CpG hypermutation is reasonably common particularly in paediatric malignancies and colorectal cancer. The authors then link somatic CpG HM events to the loss of MSH2/MSH6. Interestingly, the authors show that somatic CpG HM could be a prognostic marker for good immune checkpoint inhibitor response. Overall, the analysis is reasonably well done, but some methods are a little unconventional. I have the following suggestions for the authors to consider:

1. In the abstract, it might be more precise to say that tumour mutation burden is a biomarker for immune checkpoint inhibitors since I am not aware that it is a biomarker for any other kind of targeted therapy.
2. P8. I don't think a sample with a C>T mutations enrichment at CpG > 0.6 and a TMB above a median of just 1.33 mutations/Mb can really be considered "CpG hypermutated". The TMB threshold would need to be much higher than this, perhaps above top 10% of TMB or even higher. As it is, most of the samples being selected would likely just have a relatively strong SBS1 contribution. Would a better method be to perform mutational signature deconvolution and specifically select samples that have a very high mutation count attribution towards SBS1?
3. p.9 "We observed somatic non-CpG hypermutation in several cancer types associated with MMRd, including skin, kidney, and urinary tract cancers". I don't think skin, kidney and urinary tract cancers are really associated with MMRd.
4. Fig 2B, need to show the total CpG mutation load. Are they really hypermutated?
5. p.10 Could the mutational context of CpG mutations lead to an inflated dN/dS ratio? i.e. are C>T mutations within CpG context more likely to contribute to non-synonymous mutations compared with other mutation types?
6. p. 10 Not all cancer driver mutations are caused by point mutations. For example, some tumour suppressors can be inactivated by indels or in the case of MLH1, it is most commonly silenced epigenetically by promoter hypermethylation. Could this bias the analysis towards genes that are more likely to have driver mutations that are point mutations?
7. P12. While the MutSalph CpG hypermutation TP53 mutation is plausible, if the TP53 mutations is a later event, is its mutant allele frequency generally lower than MSH2/MSH6 mutations?
8. P13. Is it possible to check if any of the CpG hypermutators have POLE mutations? POLE mutants can have very high T[C>T]G mutation burden. In fact, in the general analysis of CpG hypermutation was there any association with POLE mutants?
9. In the discussion of the association of C>T CpG hypermutation with MutSa mutations, it would seem relevant to reference the work from Fang et al. Sci Adv 2021. This paper is cited in the introduction, but it is actually more relevant in the discussion where it provides evidence for MutSa in the repair of 5mC deamination. There is also a considerable amount of literature on the impact of CpG methylation on C>T mutations (e.g. PMID: 29223032 and 28531315).

Reviewer #3:

Flynn and colleagues describe a CpG hypermutator phenotype similar to COSMIC signature SBS1 in a cohort of ~31K samples and over 100 cancer types using an outlier detection approach supported by analysis with P-MACD. They find that the CpG hypermutator phenotype is enriched in colorectal cancer, HGGs (adult and pediatric), and pediatric leukemia. The dNdScv package was used to identify driver genes in CpG hypermutator and non-CpG hypermutator samples, with some MutS α complex genes being identified as positively selected in the CpG hypermutator group. C>T@CpG mutations in TP53 are examined. The authors identify CpG hypermutators in the MSK-IMPACT immunotherapy cohort with similar characteristics (i.e., MSH2/MSH6 mutations and in CRC) to the discovery cohort, with survival analysis results suggesting improved survival associated with CpG hypermutated phenotype. The authors propose a model where MutS α complex gene mutations lead to CpG hypermutation and precipitate somatic TP53 mutations.

This description of CpG hypermutator phenotype in a cohort of this size and breadth is likely to be of interest to cancer data

scientists. I find the methods for identifying CpG hypermutator samples adequately described and appropriate. However, key conclusions regarding the mechanistic significance of these findings are overstated, given the analyses in this version of the paper. I expect many sections would need to be heavily revised (i.e., language made more circumspect or analyses added) or removed to recommend acceptance.

Major points

1. The authors state that MutS α complex genes are the "most recurrently mutated genes in CpG hypermutators" and that there is support for a "strong genetic link between MutS α -deficiency and somatic CpG hypermutation" (pg. 10-11). However, context is missing for these claims that should be straightforward to calculate and present.

Are MutS α complex genes significantly more frequently mutated in CpG hypermutators compared to non-CpG hypermutators? This is only shown in the validation cohort (Figure 5B). What genes show deleterious mutations in 2+ CpG hypermutator samples, which appears to be the cutoff used in Figure S1C, and are other protein complexes overrepresented?

2. In the Somatic CpG mutagenesis converges on TP53-deficiency section (pg. 12), The authors demonstrate that CpG hypermutated tumors have a "three-fold higher rate of [non-synonymous] somatic C>T mutations at CpG sites" in TP53 compared to non-CpG hypermutated tumors; this, along with the fact that there's positive selection for TP53 in the driver analysis, is used as support for somatic CpG hypermutation accelerating evolution via C>T@CpG within TP53.

Classification as a CpG hypermutated tumor depends on a higher rate of somatic C>T mutations at CpG sites, so this argument has a risk of circularity. Other genes in the selection in Table S3 in the supplemental Excel sheet show a similar three-fold relationship (e.g., TDG, POLE), which suggests that there may not be anything specific about TP53 in this observation. Additionally, my interpretation of Figure 3B is that TP53 is equally likely to be a driver in CpG hypermutated and non-CpG hypermutated tumors.

I do not find this evidence sufficient to support the proposed three-step model of tumor evolution at the end of this section. Furthermore, taking into account major point 1 above, I do not think it is appropriate to conclude that the "results also suggest that MutS α -deficiency is causally linked with an accelerated somatic mutation rate at cancer driver hotspots within the TP53 tumour suppressor gene" (pg. 15).

Removing or changing the language referenced above would mitigate my concerns. The authors would need to demonstrate that the observations are unique to TP53 to leave the section on pg. 12 in. At the very least, are there any cases of longitudinal samples with deleterious mutations in MutS α complex genes and CpG hypermutation at baseline that show TP53 mutations in later samples?

3. On pg. 11: "pALL was the only cohort with recurrent PMS2 mutations." Is this true across the entire cohort or just CpG hypermutated tumors? Table S1 is referenced and only contains information for CpG hypermutated tumors.

4. Given the similarity of the CpG hypermutation signature to the SBS1 signature (Figure 1E) and the previous observation that SBS1 exposure increases with age at diagnosis (Alexandrov et al., 2015 as cited on pg. 3), I am interested in whether or not the CpG hypermutant phenotype could be explained by age in some cases. Age only appears to be taken into account in the models in Figure S4.

For samples in the dataset with age metadata available, is there a linear relationship between C>T@CpG proportion and age? If so, and one takes the residuals after regressing out age and performs the CpG outlier analysis, do the results change? It would be particularly interesting if adult samples with no deleterious alterations in MutS α complex genes were no longer CpG hypermutators after accounting for age (I would not expect this to impact the pediatric cancer results).

5. Some methods are not explained in the methods section (e.g., "official COSMIC mutational signatures (Methods)" without COSMIC mention in the Methods section) or are only mentioned in figure legends (e.g., sigfit in Figure S2). Please ensure that methods for all analyses included in figures and tables are in the text.

From a reproducibility and transparency standpoint, it would also be beneficial if code for all analyses (not only the code to compute CpG hypermutation) were made available. I understand that this is not a Molecular Systems Biology requirement.

Minor points

1. The description of the supplemental tables in the manuscript file does not match the available Excel file. I see four sheets (Table S1-S4); the manuscript PDF references six supplemental tables. This is easily resolved but impacts my ability to evaluate the manuscript.

2. I would recommend moving the panel illustrating Primary and Relapse pairs in Figure S2 (currently panel A) below Panel E to make it easier to follow with the main text primarily highlighting mutational signature exposure and genetic lesions.

3. The legend for Figure S3B references a red line, but I believe the cohort-specific 75th percentile is marked with green points.
4. I believe the last sentence on page 4 is missing a word after "'clock-like' mutational."
5. Figure 2 legend: "probably" -> "probability"

Flynn et al
Somatic CpG hypermutation is associated with mismatch repair deficiency in cancer
rebuttal

Rebuttal

Reviewer #1:

The manuscript describes analysis of somatic mutation load associated with methylated cytosines in CpG sequences (CpGTPG). This is the well-known type of motif-specific somatic mutagenesis ubiquitous in normal and in cancer tissues. Authors analyzed two cohorts: (i) a vast dataset of over 30,000 cancer samples whole-exome (WES) or whole-genome (WGS) sequenced by many different projects and belonging to multiple tumor types; (ii) and a cohort of 1,661 tumor samples in which a panel of exomes and selected introns of 341 genes were captured and sequenced (the latter cohort described in ref. (Samstein et al, 2019). There are several more conclusions of interest to cancer field.

Main conclusions of the manuscript are as follows:

A. Based on analysis of mutation calls in the 30,000-sample WES+WGS cohort authors conclude that the somatic defects in MutS alfa (MSH2-MSH6 genes) are the strongest drivers of high level of CpG mutagenesis.

B. Based on analysis of 1,661-sample cohort in which only a panel of genes (mainly exons) was sequenced authors conclude that CpG mutagenesis is a better predictor of positive response to treatment with immune checkpoint inhibitors (ICI) than overall increase in tumor mutation burden (TMB).

This is interesting study which however requires correction of a significant inaccuracy as well as addressing other comments below.

R1.1

1. Content of four Supplementary Tables presented as tabs of a single excel file "MSB-2023-12164-T-Supplementary_Tables-supp.xlsx" does not match descriptions in the text. Moreover, the text mentions six Supplementary Tables (Table S1 - Table S6). I suspect that the wrong excel file could have been uploaded at submission.

Response: We thank the reviewer for pointing out this mistake in our submission. In the revised version, we have uploaded the correct Supplementary Table file.

R1.2

2. Page 8, lines 22-24 Quote: "The CpG hypermutated tumours, on the other hand, displayed a strong similarity to SBS1 (cosine 0.95)"

Comment:

In fact, cosine similarity to SBS1 for CpG-Hi (hypermutated) tumours as shown on Fig.1E(ii) is 0.29. The values on entire panel Fig.1E(ii), which is supposed to show cosine similarities for CpG-Hi (hypermutated) tumours, are exactly the same as on

panel Fig.1E(iv), which is supposed to show cosine similarities for CpG-Lo (non-CpG hypermutated) tumours. The only difference is that panel iv is shaded in gray. Is it some kind of mixup?

Response: The table represents SBS signatures from SBS1 to SBS94 in 23 columns and 2 rows, with colour intensity representing cosine score from 0 to 1. In the revised manuscript, we have added additional description to the figure including a row-label in Figure 1E ii) and iv) ("SBS"), added an SBS cosine similarity heatmap scale at the bottom of the figure, and expanded the Figure 1 legend description as follows: *...iv) Heatmap showing the cosine similarity score (CSS, white=0, dark red=1) of the mutational signatures in i) and iii) with the COSMIC signatures, respectively. Numbers denote SBS signatures from 1 (top left) to 94 (bottom right).*"

R1.3

3, Page 6, Line 7 from bottom "using default parameters for nCg and rCg models" The rationale for the use of these two models is never explained either in the text or in the legend of Supplementary Figure 1B

Response: We have now expanded the description of the P-MACD model with an extended explanation and rationale for the nCg and rCg models in the Methods section p 6-7.

line 5 from bottom "to estimate minimum mutation loads" can be replaced for "to obtain minimum estimates of mutation loads associated with nCg-specific mutagenic mechanism" to stress that the estimate is not just a simple mutation count.

Response: Thank you for the suggestion, which we have now incorporated in the Methods section.

R1.4

4. Figure 1 and associated analysis. Mutation Np[C>T]pG motifs represent significant fraction of total mutations in melanomas, which are also often contain very high overall mutation load. However, high level of Np[C>T]pG mutagenesis in melanomas can be ascribed to UV-mutagenesis - C>T in dipyridines, TC>TT or CC>CT. Interestingly, only uveal melanomas were identified as CpG-Hi in this study. This should be explained with details of analysis in all 30000 plus samples, which is not present in the current Supplementary Table set.

Response: In response to the reviewer's question, we have performed a parallel analysis of CpG mutagenesis in both a cutaneous and uveal melanoma cohort. In the revised manuscript, we show that none of the cutaneous melanoma samples have a CpG hypermutation phenotype. In contrast, we find a single UM sample displaying the CpG hypermutation phenotype (new **Figure S2**). This is in agreement with a study from Johansson et al (PMID: 32415113), finding two subsets of UM, one with characteristic UV-mediated mutagenesis and SBS7 and the other associated with MBD4 mutations.

R1.5

5. Page 14, last para. Quote: "We find that somatic CpG hypermutation accelerates a signature resembling the 'clock-like' mutational signature SBS1, often used to time the number of cell divisions (Alexandrov et al, 2015), which suggests a replication-dependent MMR pathway, involving error-prone processing of MutSα."

Comment:

In fact, (Alexandrov et al, 2015) explain higher presence of SBS1 in tumors from tissues with high cell division rates by smaller times of T:G mismatches existence between rounds of replication, where a C>T change would be copied by DNA Pol and fixed into mutation. Only at the stretch of time a T:G mismatch exists in double-strand DNA, a thymine in the mismatch can be repaired by templated repair mechanism - BER or MMR. Importantly, in case of MMR, an unknown mechanism of strand-discrimination in non-replicating double-strand DNA should be postulated. Fang et al 2021 cited in this submission propose that error-free mismatch repair or BER of T:C mismatches created by cytidine deamination meCpG is aided by MBD4. In case of MMR, meC-specific glycosylase MBD4 can provide a tool for strand-discrimination by MMR.

Response: We thank the reviewer for this insightful comment. We modified the discussion and added a statement that MBD4 can provide strand discrimination for MMR in non-replicating dsDNA.

R1.6

6. Supplemental Figure 1. Panel B: Explain the reason of looking at rCg vs nCg.

Response: We provided more explanations in the Methods section and have updated the Figure S1 legend, referring to details about the rCg and nCg models in the Methods.

Panel C. POLE and POLD1 are catalytic large subunits of replicative DNA polymerases Pol-epsilon and Pol-delta, respectively. Their roles in MMR are not well defined. TDG and MBD4 are components of BER. Only MBD4 was shown to interact with MMR and was proposed to provide strand-discrimination for non-canonical MMR (see comment to page 14 above) This should be indicated.

Response: We thank the reviewer for this comment. We have now removed POLE and POLD1 from the Supplemental Figure 1C.

Reviewer #2:

This work by Flynn et al. represents a large scale analysis of existing cancer sequencing data focusing on somatic CpG hypermutation. The analysis suggests that somatic CpG hypermutation is reasonably common particularly in paediatric malignancies and colorectal cancer. The authors then link somatic CpG HM events to the loss of MSH2/MSH6. Interestingly, the authors show that somatic CpG HM could be a prognostic marker for good immune checkpoint inhibitor response. Overall, the analysis is reasonably well done, but some methods are a little unconventional. I have the following suggestions for the authors to consider:

R2.1

1. In the abstract, it might be more precise to say that tumour mutation burden is a biomarker for immune checkpoint inhibitors since I am not aware that it is a biomarker for any other kind of targeted therapy.

Response: The reviewer is correct, and we have made the suggested change.

R2.2

2. P8. I don't think a sample with a C>T mutations enrichment at CpG > 0.6 and a TMB above a median of just 1.33 mutations/Mb can really be considered "CpG hypermutated". The TMB threshold would need to be much higher than this, perhaps above top 10% of TMB or even higher. As it is, most of the samples being selected would likely just have a relatively strong SBS1 contribution. Would a better method be to perform mutational signature deconvolution and specifically select samples that have a very high mutation count attribution towards SBS1?

Response: We would like to clarify that we employed Tukey's fence method for defining relative mutation burden outliers for each study cohort. We reasoned that an absolute threshold for TMB outlier would be insensitive to tumour cohorts with on average low mutation burden such as several paediatric cohorts, and thereby be insensitive to mutation burden outliers with a general absolute threshold across the entire dataset. This is in agreement with other studies that have argued for a threshold above 2 to be an appropriate cut-off to identify highly mutated paediatric tumours (PMID: 36585449) To increase our stringency, we further required a minimum of 1.33 mutations per megabase. In the revised manuscript, we have expanded and clarified this point in the Methods p. 6 and Results p. 8.

R2.3

3. p.9 "We observed somatic non-CpG hypermutation in several cancer types associated with MMRd, including skin, kidney, and urinary tract cancers". I don't think skin, kidney and urinary tract cancers are really associated with MMRd.

Response: The reviewer is correct. MMRd is rare in these cancers and we have removed this sentence.

R2.4

4. Fig 2B, need to show the total CpG mutation load. Are they really hypermutated?

Response: We would like to clarify that Figures 2A and B represent summarised proportions of tissue types and cancer types with one plot showing TMB outliers and one showing CpG outliers. The actual mutation burden and CpG proportions for selected cohorts are shown in Figure 4A and Figure S3, with white dots representing 3rd quartile of the cohort and bars representing the mutation load. We acknowledge that the Figure 2 data representation could be made clearer, and we have now rephrased the description in the figure legend.

R2.5

5. p.10 Could the mutational context of CpG mutations lead to an inflated dN/dS ratio? i.e. are C>T mutations within CpG context more likely to contribute to non-synonymous mutations compared with other mutation types?

Response: Following the reviewer's question, we have computed the theoretical frequency of synonymous and nonsynonymous mutations in all possible dinucleotide contexts. We find a median proportion of 0.81 of dinucleotide mutations to cause nonsynonymous mutation, and only 0.75 for [C>T]pG mutations, showing no biased enrichment in N/S proportion for C>TpG. We have now added these results as a new Figure S1D in the revised manuscript.

R2.6

6. p. 10 Not all cancer driver mutations are caused by point mutations. For example, some tumour suppressors can be inactivated by indels or in the case of MLH1, it is most commonly silenced epigenetically by promoter hypermethylation. Could this bias the analysis towards genes that are more likely to have driver mutations that are point mutations?

Response: We include both SNVs and indels in profiling mutations of driver genes, and we have added a sentence in the revised Methods to clarify this point. We are aware of the impact of promoter hypermethylation, in particular for the *MLH1* gene. Unfortunately, DNA methylation data was not available across the cohorts. In Figure 3C, we show the results of using available mRNA expression data as a surrogate marker for promoter DNA methylation-based repression of the *MLH1* gene. Indeed, we find two cases with down-regulation of *MLH1* (z-score expression below 1.5) and a CpG hypermutation phenotype. We are aware that this likely represents an underestimate of the true number of promoter-methylated *MLH1* tumours. In Figure 3, we have 17 cases without an identified mutation in CpG hypermutated tumours, and it is likely, that a subset of these could have *MLH1* promoter hypermethylation. We note that our method to identify CpG hypermutators is independent of our ability to detect MMR mutations or aberrant promoter DNA methylation. In the revised manuscript, we have added the following sentence to highlight this limitation "*MLH1 is frequently silenced by promoter hypermethylated (Cancer Genome Atlas Network 2012), which we corroborate in several tumours using gene expression (Fig. 4). We note that this is likely to be an underestimate of the true rate of MLH1 promoter hypermethylation.*" p. 14.

R2.7

7. P12. While the MutSalphalpha CpG hypermutation TP53 mutation is plausible, if the TP53 mutations is a later event, is its mutant allele frequency generally lower than MSH2/MSH6 mutations?

Response: The reviewer raises a good point, which we have explored. We have only four cases with both TP53 and MutSAlpha mutations, which unfortunately makes a proper comparison of allele frequencies infeasible. Although we do find our hypothesis of stepwise order of events between MMR and TP53 mutations supported by our data, we do acknowledge that it will need further support in future studies. Following feedback from both Reviewers 2 and 3, we have decided to remove the proposed 3-step hypothesis.

R2.8

8. P13. Is it possible to check if any of the CpG hypermutators have POLE mutations? POLE mutants can have very high T[C>T]G mutation burden. In fact, in the general analysis of CpG hypermutation was there any association with POLE mutants?

Response: In response to the reviewer's suggestion, we investigated *POLE* hotspot mutations (P286, V411) in our study cohorts. We found no tumours with *POLE* hotspot mutations in the CpG-HM group (n=78), yet in 11 out of 1,914 TMB-Hi tumours (0.57%) and 7 out of 28,277 (0.02%) in non-hypermutated tumours. We also note that the two canonical *POLE* hotspot mutations (P286, V411) are not due to C-to-T mutations, and also do not have a CpG sequence context, suggesting that CpG mutagenesis is not causally associated with *POLE* hotspots. We have added the following sentence to the revised manuscript "*Moreover, we found 0.57% of TMB-Hi tumours to have POLE hotspot mutations, but none in CpG hypermutated tumours.*" p. 10.

R2.9

9. In the discussion of the association of C>T CpG hypermutation with MutSa mutations, it would seem relevant to reference the work from Fang et al. Sci Adv 2021. This paper is cited in the introduction, but it is actually more relevant in the discussion where it provides evidence for MutSa in the repair of 5mC deamination. There is also a considerable amount of literature on the impact of CpG methylation on C>T mutations (e.g. PMID: 29223032 and 28531315).

Response: This is a good point, and we have now inserted the following sentence and citation in the discussion of the revised manuscript: "*In support, MutSa has been demonstrated to be involved in the repair of 5mC deamination damage (Fang et al. 2021; Poulos et al. 2017; Tomkova et al. 2018)*".

Reviewer #3:

Flynn and colleagues describe a CpG hypermutator phenotype similar to COSMIC signature SBS1 in a cohort of ~31K samples and over 100 cancer types using an outlier detection approach supported by analysis with P-MACD. They find that the CpG hypermutator phenotype is enriched in colorectal cancer, HGGs (adult and pediatric), and pediatric leukemia. The dNdScv package was used to identify driver genes in CpG hypermutator and non-CpG hypermutator samples, with some MutS α complex genes being identified as positively selected in the CpG hypermutator group. C>T@CpG mutations in TP53 are examined. The authors identify CpG hypermutators in the MSK-IMPACT immunotherapy cohort with similar characteristics (i.e., MSH2/MSH6 mutations and in CRC) to the discovery cohort, with survival analysis results suggesting improved survival associated with CpG hypermutated phenotype. The authors propose a model where MutS α complex gene mutations lead to CpG hypermutation and precipitate somatic TP53 mutations.

This description of CpG hypermutator phenotype in a cohort of this size and breadth is likely to be of interest to cancer data scientists. I find the methods for identifying CpG hypermutator samples adequately described and appropriate. However, key conclusions regarding the mechanistic significance of these findings are overstated, given the analyses in this version of the paper. I expect many sections would need to be heavily revised (i.e., language made more circumspect or analyses added) or removed to recommend acceptance.

Major points

R3.1

1. The authors state that MutS α complex genes are the "most recurrently mutated genes in CpG hypermutators" and that there is support for a "strong genetic link between MutS α -deficiency and somatic CpG hypermutation" (pg. 10-11). However, context is missing for these claims that should be straightforward to calculate and present.

Are MutS α complex genes significantly more frequently mutated in CpG hypermutators compared to non-CpG hypermutators? This is only shown in the validation cohort (Figure 5B). What genes show deleterious mutations in 2+ CpG hypermutator samples, which appears to be the cutoff used in Figure S1C, and are other protein complexes overrepresented?

Response: We have followed the reviewer's advice and computed enrichment statistics on MMR mutations for the cohort, which is in agreement with our enrichment analysis in the ICI cohort (Figure 5B). We have now added the following analyses to the revised manuscript: "[...] mutations in the MutS α complex (MSH2/MSH6) were enriched in CpG-HM compared to both TMB-high (OR=2.5, P=0.001) and TMB-low tumours (OR=48, P<2.2e-16) [...]". p. 10.

R3.2

2. In the Somatic CpG mutagenesis converges on TP53-deficiency section (pg. 12), The authors demonstrate that CpG hypermutated tumors have a "three-fold higher rate of

[non-synonymous] somatic C>T mutations at CpG sites" in TP53 compared to non-CpG hypermutated tumors; this, along with the fact that there's positive selection for TP53 in the driver analysis, is used as support for somatic CpG hypermutation accelerating evolution via C>T@CpG within TP53.

Classification as a CpG hypermutated tumor depends on a higher rate of somatic C>T mutations at CpG sites, so this argument has a risk of circularity. Other genes in the selection in Table S3 in the supplemental Excel sheet show a similar three-fold relationship (e.g., TDG, POLE), which suggests that there may not be anything specific about TP53 in this observation. Additionally, my interpretation of Figure 3B is that TP53 is equally likely to be a driver in CpG hypermutated and non-CpG hypermutated tumors.

I do not find this evidence sufficient to support the proposed three-step model of tumor evolution at the end of this section. Furthermore, taking into account major point 1 above, I do not think it is appropriate to conclude that the "results also suggest that MutS α -deficiency is causally linked with an accelerated somatic mutation rate at cancer driver hotspots within the TP53 tumour suppressor gene" (pg. 15).

Removing or changing the language referenced above would mitigate my concerns. The authors would need to demonstrate that the observations are unique to TP53 to leave the section on pg. 12 in. At the very least, are there any cases of longitudinal samples with deleterious mutations in MutS α complex genes and CpG hypermutation at baseline that show TP53 mutations in later samples?

Response: The reviewer raises an important point. Due to the low number of co-occurring TP53 and MutS α cases (N=4), and the associated challenges in further supporting our model, we have decided to follow the reviewer's recommendation and removed the suggested stepwise model in the revised manuscript (see also comment to R2.7).

R3.3

3. On pg. 11: "pALL was the only cohort with recurrent PMS2 mutations." Is this true across the entire cohort or just CpG hypermutated tumors? Table S1 is referenced and only contains information for CpG hypermutated tumors.

Response: The usage of the word recurrence was imprecise, and we have therefore removed this sentence.

R3.4

4. Given the similarity of the CpG hypermutation signature to the SBS1 signature (Figure 1E) and the previous observation that SBS1 exposure increases with age at diagnosis (Alexandrov et al., 2015 as cited on pg. 3), I am interested in whether or not the CpG hypermutant phenotype could be explained by age in some cases. Age only appears to be taken into account in the models in Figure S4.

For samples in the dataset with age metadata available, is there a linear relationship between C>T@CpG proportion and age? If so, and one takes the residuals after regressing out age and performs the CpG outlier analysis, do the results change? It would be particularly interesting if adult samples with no deleterious alterations in

MutS α complex genes were no longer CpG hypermutators after accounting for age (I would not expect this to impact the pediatric cancer results).

Response: The reviewer proposes a mutagenic process by which C-to-T hypermutations in CpG context accumulate as a function of patient age in a clock-like manner. To investigate this hypothesis, we performed a linear regression of mutations and age, which showed a very weak linear relationship for C>T@CpG proportion and age ($R=0.11$, $p\text{-value}<2.2e-16$), and a non-significant negative linear relationship in CpG-HM samples ($R=-0.14$, $p\text{-value}=0.3$). This suggests that somatic CpG hypermutation is not clock-like, but more likely occurring over a shorter time scale, likely following a second hit in BER/MMR genes.

R3.5

5. Some methods are not explained in the methods section (e.g., "official COSMIC mutational signatures (Methods)" without COSMIC mention in the Methods section) or are only mentioned in figure legends (e.g., sigfit in Figure S2). Please ensure that methods for all analyses included in figures and tables are in the text.

From a reproducibility and transparency standpoint, it would also be beneficial if code for all analyses (not only the code to compute CpG hypermutation) were made available. I understand that this is not a Molecular Systems Biology requirement.

Response: We thank the reviewer for the suggestion, and we have now made a git repository available (https://bitbucket.org/weischenfeldt/cpghm_publication_code).

R3.6

Minor points

1. The description of the supplemental tables in the manuscript file does not match the available Excel file. I see four sheets (Table S1-S4); the manuscript PDF references six supplemental tables. This is easily resolved but impacts my ability to evaluate the manuscript.

Response: We apologise for this mistake in uploading the supplemental files and have corrected this in the revised version.

R3.7

2. I would recommend moving the panel illustrating Primary and Relapse pairs in Figure S2 (currently panel A) below Panel E to make it easier to follow with the main text primarily highlighting mutational signature exposure and genetic lesions.

Response: We have followed the reviewer's suggestion and moved the panel to the lower part of Figure S3.

R3.8

3. The legend for Figure S3B references a red line, but I believe the cohort-specific 75th percentile is marked with green points.

Response: We assume the reviewer is referring to a mistake in the former Figure S2B (new Figure S3B), which we have corrected in the revised manuscript.

R3.9

4. I believe the last sentence on page 4 is missing a word after "'clock-like' mutational."

Response: Thank you for pointing out this error, which we have corrected to "*understanding the 'clock-like' mutational signature*"

R3.10

5. Figure 2 legend: "probably" -> "probability"

Response: We have corrected this mistake.

23rd May 2024

Manuscript Number: MSB-2023-12164R

Title: Somatic CpG hypermutation is associated with mismatch repair deficiency in cancer

Dear Prof Weischenfeldt,

Thank you for the submission of your revised manuscript to Molecular Systems Biology. We have now received the enclosed reports from the referees that were asked to re-assess it. As you will see the reviewers are now globally supportive and I am pleased to inform you that we will be able to accept your manuscript pending the following final amendments:

1) In the main manuscript file, please include keywords to max. 5.

2) Please move the information from the 'Supplemental Code' section into the Data availability section and format according to the example below:

"The datasets and computer code produced in this study are available in the following databases:

- Modeling computer scripts: GitHub (<https://github.com/SysBioChalmers/GECKO/releases/tag/v1.0>)

- [data type]: [full name of the resource] [accession number/identifier] ([doi or URL or identifiers.org/DATABASE:ACCESSION])"

3) Please rename "Conflict of Interest statement" to "Disclosure and competing interests statement". We updated our journal's competing interests policy in January 2022 and request authors to consider both actual and perceived competing interests. Please review the policy <https://www.embopress.org/competing-interests> and update your competing interests if necessary.

4) Author contributions: Please remove it from the manuscript and specify author contributions in our submission system. CRediT has replaced the traditional author contributions section because it offers a systematic machine-readable author contributions format that allows for more effective research assessment. You are encouraged to use the free text boxes beneath each contributing author's name to add specific details on the author's contribution. More information is available in our guide to authors:

<https://www.embopress.org/page/journal/17574684/authorguide#authorshipguidelines>

5) Data citations: Data citations and references should include links to specific accession codes used in the study, not to a general database (e.g. a generic URL is provided for GLASS Consortium Synapse repository, PedcBioPortal data citation, etc). In studies that make use of many (i.e. more than approx. 20) pre-existing datasets as in this case, we realize it is not practical or feasible to cite them individually in the main manuscript. In this case, it is therefore acceptable to provide a separate data reference list in the form of an Expanded View Table which should be called out from Materials & Methods. Further instructions are available at .

6) Data not shown: We do not allow statements/conclusions with "data not shown". As per our guidelines, on "Unpublished Data" the journal does not permit citation of "Data not shown". All data referred to in the paper should be displayed in the main or Expanded View figures. Please remove from page 22.

7) In the Materials and Methods, please ensure that a statement on whether or not blinding was done is included in the Materials and Methods even if no blinding was done. Please also update the Author Checklist with this information.

8) Please place individual sections of the manuscript in the following order: Title page - Abstract & Keywords - Introduction - Results - Discussion - Materials & Methods - Data Availability - Acknowledgements - Disclosure and Competing Interests Statement - References - Figure Legends - Expanded View Figure Legends.

9) For the figures and figure legends, please take care of the following:

- Please make sure to update the callouts of all figures in the main manuscript text. Expanded View figures should be named and labeled correctly as Figure EV1-EV5 throughout the manuscript.

- Please define the annotated p values *** as well as provide the exact p-value for the same in the legend of figure 5e; as appropriate.

- Please note that the exact p values are not provided in the legends of figures 5b, d; EV 5a.

- Please indicate the statistical test used for data analysis in the legends of figures 3b; 5e.

- Please note that the box plots need to be defined in terms of minima, maxima, in the legend of figure 1b.

- Please note that the box plots need to be defined in terms of minima, maxima, centre, bounds of box and whiskers, and percentile in the legends of figures 1c; 5e; EV 1a; EV 2a-b.

- Please note that information related to n is missing in the legends of figures 3a; 5e; EV 2a-b.

- Although 'n' is provided, please describe the nature of entity for 'n' in the legend of figure EV 1a.

- Please note that the error bars are not defined in the legends of figures 3a; EV 5a-b.

10) Funding: Please note that funding information should be given in the "Acknowledgements" section (not in its own separate section).

11) Synopsis:

- Synopsis image: Please provide a graphic that summarises the main findings of the manuscript on a glance and upload it as a high-resolution jpeg file 550 pixels wide x (250-400) pixels high.

- Synopsis text: Please provide a short standfirst (maximum of 300 characters, including space), limit the bullet points to max. 5 and upload it as a separate .doc file. Please write the bullet points to summarise the key NEW findings. They should be designed to be complementary to the abstract - i.e. not repeat the same text. We encourage inclusion of key acronyms and quantitative information (maximum of 30 words / bullet point). Please use the passive voice and upload this as a separate file.

12) As part of the EMBO Publications transparent editorial process initiative (see our policy here: https://www.embopress.org/transparent-process#Review_Process), Molecular Systems Biology will publish online a Peer Review File (PRF) to accompany accepted manuscripts. This file will be published in conjunction with your paper and will include the anonymous referee reports, your point-by-point response and all pertinent correspondence relating to the manuscript. Let us know whether you agree with the publication of the PRF and as here, if you want to remove or not any figures from it prior to publication. Please note that the Authors checklist will be published at the end of the PRF.

13) Please provide a point-by-point letter INCLUDING my comments as well as the reviewer's reports and your detailed responses (as Word file).

I look forward to reading a new revised version of your manuscript as soon as possible.

Yours sincerely,

Poonam Bheda, PhD
Scientific Editor
Molecular Systems Biology

Reviewer #1:

Authors did commendable large job to address my as well as other reviewers comments. I recommend acceptance.

Reviewer #2:

The authors have addressed all my concerns.

***While Reviewer 3 was not able to re-review, both Reviewers 1 and 2 discussed with me in a cross-commenting session that your responses to the Reviewer 3's previous concerns have been sufficiently addressed as well.

Flynn et al
Somatic CpG hypermutation is associated with mismatch repair deficiency in cancer
Point-by-point response

Manuscript Number: MSB-2023-12164R

Title: Somatic CpG hypermutation is associated with mismatch repair deficiency in cancer

Point-by-point response

- 1) In the main manuscript file, please include keywords to max. 5.
- 2) Please move the information from the 'Supplemental Code' section into the Data availability section and format according to the example below:

"The datasets and computer code produced in this study are available in the following databases:

- Modeling computer scripts: GitHub

(<https://github.com/SysBioChalmers/GECKO/releases/tag/v1.0>)

- [data type]: [full name of the resource] [accession number/identifier] ([doi or URL or identifiers.org/DATABASE:ACCESSION)]"

Response:

We have included five keywords and moved the information in the supplemental code section to the Data Availability section

- 3) Please rename "Conflict of Interest statement" to "Disclosure and competing interests statement". We updated our journal's competing interests policy in January 2022 and request authors to consider both actual and perceived competing interests. Please review the policy <https://www.embopress.org/competing-interests> and update your competing interests if necessary.

Response:

We have renamed the "Conflict of Interest statement" section to "Disclosure and competing interests statement" accordingly.

- 4) Author contributions: Please remove it from the manuscript and specify author contributions in our submission system. CRediT has replaced the traditional author contributions section because it offers a systematic machine-readable author contributions format that allows for more effective research assessment. You are encouraged to use the free text boxes beneath each contributing author's name to add specific details on the author's contribution. More information is available in our guide to authors:

<https://www.embopress.org/page/journal/17574684/authorguide#authorshipguidelines>

Response:

We have removed the author's contributions from the manuscript and added it to the submission system

5) Data citations: Data citations and references should include links to specific accession codes used in the study, not to a general database (e.g. a generic URL is provided for GLASS Consortium Synapse repository, PedcBioPortal data citation, etc). In studies that make use of many (i.e. more than approx. 20) pre-existing datasets as in this case, we realize it is not practical or feasible to cite them individually in the main manuscript. In this case, it is therefore acceptable to provide a separate data reference list in the form of an Expanded View Table which should be called out from Materials & Methods. Further instructions are available at <https://www.embopress.org/page/journal/17574684/authorguide#referencesformat>.

Response:

We have now generated a separate data reference list in an Expanded View Table 6, which we reference in the Materials and Methods section.

6) Data not shown: We do not allow statements/conclusions with "data not shown". As per our guidelines, on "Unpublished Data" the journal does not permit citation of "Data not shown". All data referred to in the paper should be displayed in the main or Expanded View figures. Please remove from page 22.

Response:

We have removed the sentence and instead refer to the data

7) In the Materials and Methods, please ensure that a statement on whether or not blinding was done is included in the Materials and Methods even if no blinding was done. Please also update the Author Checklist with this information.

Response:

We have added a sentence to the Materials and Methods section and updated the Author Checklist accordingly

8) Please place individual sections of the manuscript in the following order: Title page - Abstract & Keywords - Introduction - Results - Discussion - Materials & Methods - Data Availability - Acknowledgements - Disclosure and Competing Interests Statement - References - Figure Legends - Expanded View Figure Legends.

Response:

The manuscript sections have now been updated with the correct ordering

9) For the figures and figure legends, please take care of the following:

- Please make sure to update the callouts of all figures in the main manuscript text. Expanded View figures should be named and labeled correctly as Figure EV1-EV5 throughout the manuscript.

Response:

We have checked that figures are called out and checked all expanded view labels throughout the manuscript

- Please define the annotated p values *** as well as provide the exact p-value for the same in the legend of figure 5e; as appropriate.

Response:

We now define the p-value asterisks and include the exact p-value for Figure 5e in the legend.

- Please note that the exact p values are not provided in the legends of figures 5b, d; EV 5a.

Response:

We now provide exact p-values in the legends of Figure 5 and EV5.

- Please indicate the statistical test used for data analysis in the legends of figures 3b; 5e.

Response:

We now provide the statistical tests used in the analyses of data for Figure 3b and 5e.

- Please note that the box plots need to be defined in terms of minima, maxima, in the legend of figure 1b.

Response:

Boxplots are now defined in the legend of Figure 1

- Please note that the box plots need to be defined in terms of minima, maxima, centre, bounds of box and whiskers, and percentile in the legends of figures 1c; 5e; EV 1a; EV 2a-b.

Response:

Boxplots are now defined in the legend of Figure 1c, 5e, EV1a and EV2ab

- Please note that information related to n is missing in the legends of figures 3a; 5e; EV 2a-b.

Response:

The number of samples used is now provided in the legends of Figure 3a, 5e and EV2ab.

- Although 'n' is provided, please describe the nature of entity for 'n' in the legend of figure EV 1a.

Response:

The cancer mutation type is now defined for EV1a

- Please note that the error bars are not defined in the legends of figures 3a; EV 5a-b.

Response:

We now define the error bars in the legends of Figure 3a and EV5ab.

10) Funding: Please note that funding information should be given in the "Acknowledgements" section (not in its own separate section).

Response:

We have moved the funding information to the Acknowledgements section.

11) Synopsis:

- Synopsis image: Please provide a graphic that summarises the main findings of the manuscript on a glance and upload it as a high-resolution jpeg file 550 pixels wide x (250-400) pixels high.

Response:

A synopsis image is provided as an uploaded image

- Synopsis text: Please provide a short standfirst (maximum of 300 characters, including space), limit the bullet points to max. 5 and upload it as a separate .doc file. Please write the bullet points to summarise the key NEW findings. They should be designed to be complementary to the abstract - i.e. not repeat the same text. We encourage inclusion of key acronyms and quantitative information (maximum of 30 words / bullet point). Please use the passive voice and upload this as a separate file.

Response:

A synopsis text is provided separately.

12) As part of the EMBO Publications transparent editorial process initiative (see our policy here: https://www.embopress.org/transparent-process#Review_Process), Molecular Systems Biology will publish online a Peer Review File (PRF) to accompany accepted manuscripts. This file will be published in conjunction with your paper and will include the anonymous referee reports, your point-by-point response and all pertinent correspondence relating to the manuscript. Let us know whether you agree with the publication of the PRF and as here, if you want to remove or not any figures from it prior to publication. Please note that the Authors checklist will be published at the end of the PRF.

Response:

We have checked the text and we approve to the publication of a PRF.

13) Please provide a point-by-point letter INCLUDING my comments as well as the reviewer's reports and your detailed responses (as Word file).

Reviewer #1:

Authors did commendable large job to address my as well as other reviewers comments. I recommend acceptance.

Response:

We thank the reviewer for all the constructive feedback and for recommending acceptance!

Reviewer #2:

The authors have addressed all my concerns.

Response:

We thank the reviewer for all the constructive feedback and for recommending acceptance!

***While Reviewer 3 was not able to re-review, both Reviewers 1 and 2 discussed with me in a cross-commenting session that your responses to the Reviewer 3's previous concerns have been sufficiently addressed as well.

28th Jun 2024

Manuscript number: MSB-2023-12164RR

Title: Somatic CpG hypermutation is associated with mismatch repair deficiency in cancer

Dear Prof Weischenfeldt,

Thank you again for sending us your revised manuscript. We are now satisfied with the modifications made and I am pleased to inform you that your paper has been accepted for publication.

Yours sincerely,

Sincerely,

Poonam Bheda, PhD
Scientific Editor
Molecular Systems Biology
